# Towards reconstructing the Arctic atmospheric methane history over the 20th century: measurement and modeling results for the NGRIP firn

Taku Umezawa[1], Satoshi Sugawara[2], Kenji Kawamura[3,4,5], Ikumi Oyabu[3], Stephen J. Andrews[1,6], Takuya Saito[1], Shuji Aoki[7] and Takakiyo Nakazawa[7]

[1]National Institute for Environmental Studies, Tsukuba, Japan
[2]Miyagi University of Education, Sendai, Japan
[3]National Institute of Polar Research, Tokyo, Japan
[4]Department of Polar Science, The Graduate University of Advanced Studies (SOKENDAI), Tokyo, Japan
[5]Japan Agency for Marine Science and Technology, Yokosuka, Japan
[6]Wolfson Atmospheric Chemistry Laboratories, Department of Chemistry, University of York, York, UK
[7]Center for Atmospheric and Oceanic Studies, Graduate School of Science, Tohoku University, Sendai, Japan

*Correspondence to*: Taku Umezawa (umezawa.taku@nies.go.jp)

**Abstract.** Systematic measurements of atmospheric methane ($CH_4$) mole fractions at the northern high latitudes only began in the early 1980s. Although $CH_4$ measurements from Greenland ice cores consistently covered the period before ~1900, no reliable observational record is available for the intermediate period. We newly report a data set of trace gases from the air trapped in firn (an intermediate stage between snow and glacial ice formation) collected at the NGRIP (North Greenland Ice Core Project) site in 2001. We also use a set of published firn-air data at the NEEM (North Greenland Eemian ice Drilling) site. The two Arctic firn air data sets are analyzed with a firn-air transport model, which translates historical variations to depth profiles of trace gases in firn. We examine a variety of possible firn diffusivity profiles using a suite of measured trace gases, and reconstruct the $CH_4$ mole fraction by an iterative dating method. Although the reconstructions of the Arctic $CH_4$ mole fraction before the mid-1970s still has large uncertainties (> 30 ppb), we find a relatively narrow range of atmospheric $CH_4$ history that is consistent with both depth profiles of NGRIP and NEEM. The atmospheric $CH_4$ history inferred by this study is more consistent with the atmospheric $CH_4$ scenario prepared for the NEEM firn modeling than that for the CMIP6 (Climate Model Intercomparison Project Phase 6) experiments. Our study shows that the atmospheric $CH_4$ scenario used for the NEEM firn modeling is considered to be the current best choice for the Arctic $CH_4$ history, but it should not be used to tune firn-air transport models until verified by further measurements from sources such as Arctic ice cores. Given the current difficulty in reconstructing the $CH_4$ history with low uncertainty from the firn-air data sets from Greenland, future sampling and measurements of ice cores at a high-accumulation site may be the only way to accurately reconstruct the atmospheric $CH_4$ trend over the 20th century.

# 1 Introduction

Methane ($CH_4$) is an important atmospheric greenhouse gas emitted from both natural and anthropogenic sources. Despite great efforts for understanding its global budget, emission estimates of individual sources still have large quantitative uncertainties (e.g. Saunois et al., 2020; Chandra et al., 2021). Anthropogenic activities have enhanced $CH_4$ emissions globally and more than doubled the abundance of atmospheric $CH_4$ over the industrial era (e.g. Etheridge et al., 1998). The $CH_4$ emission histories have been estimated based on human activity statistics combined with emission factors (Stern and Kaufmann 1996; van Aardenne et al., 2001). Such historical emission inventories have been examined by atmospheric chemistry transport modeling (Houweling et al., 2000; Monteil et al., 2011; Ghosh et al., 2015), in combination with the records of atmospheric $CH_4$ mole fraction reconstructed from polar ice cores (Blunier et al., 1993; Nakazawa et al., 1993; Etheridge et al., 1998; MacFarling Meure et al., 2006; Sapart et al., 2012) and air extracted from porous snow layers at the top of ice sheets (firn) (Francey et al., 1999; Buizert et al., 2012; Sapart et al., 2013).

A large fraction of natural and anthropogenic $CH_4$ sources resides in the northern hemisphere, and thus the atmospheric $CH_4$ trend of the northern hemisphere can provide important information on the evolution of anthropogenic $CH_4$ emissions as well as the variations of natural $CH_4$ emission in response to climatic variability. The interhemispheric gradient of $CH_4$ mole fraction is also key to the allocation of $CH_4$ emissions between both hemispheres (e.g. Dlugokencky et al., 2003; Ghosh et al., 2015; Chandra et al., 2021).

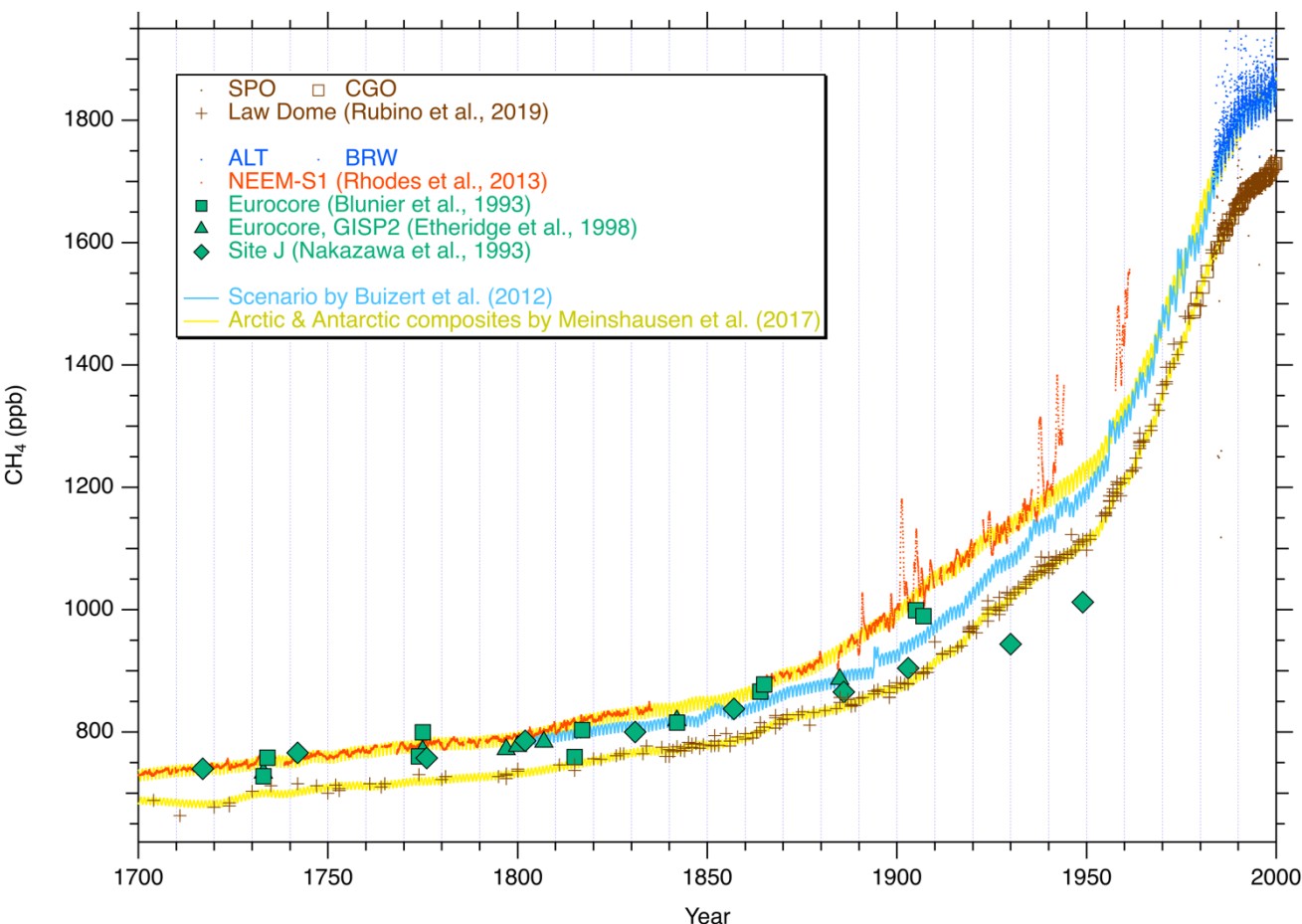

**Figure 1: Atmospheric CH₄ mole fraction data covering the last 300 years. Symbols in brown are from the southern hemisphere; crosses and open circles are from the Law Dome ice core and firn, respectively (Etheridge et al., 1998; MacFarling Meure et al., 2006; Rubino et al., 2019); open squares are from Cape Grim (CGO) (MacFarling Meure et al., 2006); dots are from South Pole (SPO) ([ftp://aftp.cmdl.noaa.gov/](ftp://aftp.cmdl.noaa.gov/)). Other colored data (except yellow) from the northern hemisphere; closed green squares, triangles and diamonds are from the Eurocore (Blunier et al., 1993), Eurocore and GISP2 (Etheridge et al., 1998) and Site J (Nakazawa et al., 1993) ice cores; red dots are the NEEM-S1 ice core (Rhodes et al., 2013); blue dots are from Alert, Canada (ALT) and Barrow, Alaska (BRW) ([ftp://aftp.cmdl.noaa.gov/](ftp://aftp.cmdl.noaa.gov/)); light blue and yellow lines are the atmospheric scenarios prepared for the NEEM firn modeling (Buizert et al., 2012) and the CMIP6 experiments (Meinshausen et al. 2017), respectively. For the latter, Arctic and Antarctic scenarios respectively with higher and lower mole fractions are shown.**

Systematic measurements of atmospheric CH₄ mole fraction began in the 1980s. In Figure 1, CH₄ measurements at two Arctic sites: Barrow, Alaska (BRW) and Alert, Canada (ALT) are shown (blue dots), whose records start in 1983 and 1985, respectively (data provided by National Oceanic and Atmospheric Administration/Earth System Research Laboratory/Global Monitoring Laboratory, NOAA/ESRL/GML). Although some sparse data from the 1970s are available (e.g. Rice et al., 2016), such "direct" measurements provide CH₄ data only since around 1980, which means that some reconstruction methodology is required to infer atmospheric CH₄ mole fraction variations before that time. For this purpose, air extracted from ice cores and

firn layers have been measured. Figure 1 also presents $CH_4$ mole fractions analyzed in ice cores from Greenland (closed green symbols), such as Eurocore (Blunier et al., 1993), Eurocore and GISP2 (Greenland Ice Sheet Project 2) (Etheridge et al., 1998) and Site J (Nakazawa et al., 1993). These data show fairly good agreements with each other until ~1900, after which the number of data and the consistency among the records are poor. Continuous measurements from NEEM-S1 ice core (Rhodes et al., 2013) presented $CH_4$ mole fractions before ~1960 (red in Figure 1), but their data are notably higher than other ice core

data after ~1850 (Blunier et al., 1993; Etheridge et al., 1998; Nakazawa et al., 1993). Therefore, the inconsistency among the different data sets indicates that the period between ~1900 and ~1980 is not covered reliably either by direct observations or ice core reconstructions.

In comparison to these data, higher-resolution $CH_4$ data are available from Antarctica (brown in Figure 1). The comprehensive Law Dome ice core and firn data set (Etheridge et al., 1998; MacFarling Meure et al., 2006; Rubino et al., 2019) almost

continuously covers the last 300 years and are well connected to the direct measurements at South Pole (SPO, data provided by NOAA/ESRL/GML) and Cape Grim (CGO) (Etheridge et al., 1998; MacFarling Meure et al., 2006). Such consistency and continuity among the datasets suggest that the Antarctic $CH_4$ data can serve as a good reference to represent the global atmospheric $CH_4$ trend over the past centuries.

Despite the limited observational data as described above, two synthetic data sets for the Arctic historical $CH_4$ mole fractions

are currently available (see section 3.2 for details). Buizert et al., (2012) prepared Arctic historical trends of mole fractions of atmospheric trace gases including $CH_4$ (light blue in Figure 1), which in turn was used to constrain the gas diffusivity profile in firn at the NEEM site. Considered as the most likely atmospheric $CH_4$ trend for the northern high latitudes, this scenario was treated as a "known" history, by which the diffusivity profiles in firn were tuned in firn-air transport models (Witrant et al., 2012; Trudinger et al., 2013). The other is the composite data set prepared for use in the CMIP6 experiments (yellow in

Figure 1, Meinshausen et al. 2017). It is seen that their scenario for the northernmost latitude (higher-mole fraction yellow line in Figure 1) follows the NEEM-S1 ice core data set (red) and inconsistent with the Buizert et al. (2012) scenario (light blue).

For constraining the global $CH_4$ budget, isotope ratios of $CH_4$ could provide useful information for relative contributions of different types of $CH_4$ sources. Sapart et al. (2013) examined the reconstruction of stable carbon isotope ratio ($\delta^{13}C$) of atmospheric $CH_4$ using firn-air measurements from both northern and southern hemispheres. They concluded that, with the

available firn measurements and understanding of firn-air transport, it is difficult to consistently reconstruct the past trend of $\delta^{13}C$ of $CH_4$ because of multiple reasons including uncertainty in the atmospheric $CH_4$ mole fraction scenario. Among many important and uncertain factors, the accurate reconstruction of the atmospheric $CH_4$ mole fraction is particularly important, because the trend in the mole fraction can lead to significant signal in the modeled $\delta^{13}C$ profile in firn due to the difference in the molecular diffusion coefficient, even in the absence of a temporal trend in atmospheric $\delta^{13}C$. Reducing uncertainty of the

historical trend of the $CH_4$ mole fraction is therefore important also for utilizing isotope data for better understanding of historical changes of different $CH_4$ emission categories.

In this study, we present a set of mole fractions of $CH_4$ and other trace gases in firn-air samples collected at the NGRIP site. Using the available atmospheric scenarios, we simulate the depth profiles of trace gases in the NGRIP firn with our firn-air

transport model as well as those in the NEEM firn reported previously (Buizert et al., 2012). It will be shown in series of firn
modeling in this study that, compared to other trace gases, $CH_4$ is uniquely underconstrained in its Arctic history. We examine
a variety of modeling cases for different diffusivity profiles and reconstruct the Arctic atmospheric $CH_4$ over the late 20th
century using an iterative dating approach (Trudinger et al., 2002). The reconstructed $CH_4$ trends from both firn are evaluated
by comparison to the atmospheric $CH_4$ scenarios. Uncertainty of the Arctic atmospheric $CH_4$ history for use in firn-air modeling
is discussed.

## 2 Experimental method

Firn air was sampled at the Greenland site NGRIP (75.10° N, 42.32° W, 2959 m AMSL) in May–June 2001. Mean
accumulation, surface density, temperature and pressure are 179 kg m$^{-2}$ yr$^{-1}$, 300 kg m$^{-3}$, 241 K and 680 hPa, respectively.
Details of the firn and firn-air sampling have been described elsewhere (Kawamura et al., 2006; Ishijima et al., 2007). At the
NGRIP site, two shallow holes (EU and Japanese holes) were drilled (Kawamura et al., 2006; Landais et al., 2006), and the
present data are from the firn-air samples collected from the Japanese hole. The total number of air-sampling depths is 24.
Since the technical details are reported in Kawamura et al. (2021), only brief descriptions of relevant data presented in this
study are given here. $CH_4$ mole fractions of the firn-air samples were measured using a gas chromatograph (Agilent 6890,
Agilent Technologies Inc.) equipped with a flame ionization detector (GC-FID) at Tohoku University (TU), with a
reproducibility of 2 ppb (Umezawa et al., 2014). The $CH_4$ mole fractions were determined against our working standard gases
that were calibrated on the TU1987 $CH_4$ scale (Aoki et al., 1992; Umezawa et al., 2014; Fujita et al., 2018). The difference
between the TU1987 $CH_4$ scale and the WMO $CH_4$ mole fraction scale (on which the NEEM $CH_4$ data were measured) is
estimated to be ~0.5 ppb at the current atmospheric $CH_4$ levels (Fujita et al., 2018). Oyabu et al. (2020) reported that ice core
data analyzed on the TU1987 and WMO scales showed good agreement within analytical uncertainties, indicating consistency
of both scales, including for the lower mole fractions (e.g. ~700 ppb). It is therefore likely that the difference between both
scales is well below the variations of interest in this study, and thus no correction is applied for use of the NGRIP and NEEM
firn data.
The firn-air samples were measured for $CO_2$ and $SF_6$ mole fractions respectively by using a nondispersive infrared gas analyzer
(NDIR) and a gas chromatograph equipped with an electron capture detector (GC-ECD) at TU. The measurement
reproducibility is estimated to be 0.02 ppm for $CO_2$ and 0.09 ppt for $SF_6$ and mole fractions of both gases are reported on the
TU2010 $CO_2$ and TU2002 $SF_6$ scales, respectively (Sugawara et al., 2018). Although not presented in this study, the firn-air
samples were previously analyzed also for nitrous oxide ($N_2O$) and its isotope ratios (Ishijima et al., 2007).
As part of this study, the NGRIP firn-air samples were newly analyzed for selected halocarbons (CFC-11, CFC-12, CFC-113
and $CH_3CCl_3$) on the Vacuum Preconcentration and Refocusing-Gas Chromatography-Mass Spectrometry (VPR-GCMS)
system, which was developed based on the work by Saito et al. (2006). An aliquot of the sample was transferred into an
evacuated canister of ~0.3 L at around ambient pressure (~100 kPa) and the inner pressure of the canister was recorded. The

air is extracted by a vacuum pump through a preconcentration trap filled with HayeSep D cooled to –135° C using a Stirling cooler. The preconcentration trap was heated to –70°C to release major atmospheric constituents and then up to 100°C to transfer the trapped compounds to a cryofocusing trap containing Carboxene 1000/Tenax TA at −100°C. The trap was then heated to 180°C to inject the trapped gases onto a PoraBOND Q separation column for subsequent analysis on MS. Mole
fractions of individual halocarbons are determined against a working standard gas (compressed dry air) that was calibrated against synthetic standards on the NIES-08 scales.

We also use a suit of trace gas measurement data from the NEEM site (77.45° N, 51.06° W, 2450 m AMSL). The firn air samples were collected in July 2008. The details of firn air sampling and gas measurements have been described by Buizert et al. (2012). The depth profile data of all the above trace gases ($CH_4$, $CO_2$, $SF_6$, CFC-11, CFC-12, CFC-113 and $CH_3CCl_3$) from
the NEEM firn are used in this study as well as HFC-134a and $^{14}CO_2$ data that are available from NEEM but not for NGRIP.

**3 Firn-air transport model**

Since the gas diffusivity in firn layers is significantly lower than in the atmosphere, the movements of atmospheric constituents are driven mostly by molecular diffusion according to their vertical mole fraction gradients under the influence of gravity. In general, lighter air components (or isotopologues) diffuse faster under their mole fraction gradients, while heavier components
accumulate in the deeper layers due to the gravitational effect. Hence the depth profiles of trace gas mole fractions in firn are determined by the atmospheric histories transferred towards depth in the firn by the molecular diffusion driven by the mole fraction gradient and gravity. At the bottom of firn, the air is trapped as bubbles in the ice sheet, which creates slow downward motion of firn air.

The firn column can be divided into three zones: a convective zone (CZ), a diffusive zone (DZ) and a lock-in zone (LIZ)
(Sowers et al., 1992; Kawamura et al., 2006; Buizert et al., 2012). In CZ, primarily driven by surface winds and fluctuations of atmospheric pressure, air is mixed with the overlying atmosphere (Sowers et al., 1992; Kawamura et al., 2006). The CZ thickness is estimated to be below 2 m at NGRIP (Kawamura et al., 2006) and 4.5 m at NEEM. In DZ, which is sufficiently isolated from the surface turbulence, movement of air is governed by molecular diffusion. Gravitational enrichment according to the barometric equation (i.e. linear increases of $\delta^{15}N$ of $N_2$ and $\delta^{18}O$ of $O_2$) occurs with depth and stops at the top of LIZ
(Sowers et al., 1992; Schwander et al., 1993; Kawamura et al., 2006). The top of LIZ (lock-in depth) is at depth 63 m coincidently same at NGRIP (Kawamura et al., 2006) and NEEM (Buizert et al., 2012). In LIZ, advection with the enclosing ice matrix dominates the transport of air, and air parcels are gradually isolated as bubbles. Traditionally, it was supposed that high-density impermeable ice layers stop diffusivity in LIZ completely, however, recent studies demonstrated finite diffusivity in LIZ (Severinghaus et al., 2010; Buizert et al., 2012; Trudinger et al., 2013). The deepest air sampling was successfully made
at 77.71 m at NGRIP (Kawamura et al., 2021) and 77.75 m at NEEM (Buizert et al., 2012), and total pore closure is considered in the deeper layers in our modeling.

## 3.1 Modeling firn-air transport

We use a one-dimensional diffusion model that has been used for the reconstruction of isotope ratios of $CO_2$ and $N_2O$ (Sugawara et al., 2003; Ishijima et al., 2007). The model is conceptually similar to that developed by Trudinger et al. (1997); it is based on a theoretical formation of diffusion (Schwander et al., 1993) and a bubble trapping process (Rommelaere et al., 1997). Air movement in the firn is driven by molecular diffusion and a gravitational effect. Namely, a trace gas flux ($F$) in firn is expressed by

$$F = -D \left\{ s \frac{\partial}{\partial z} \left( \frac{c}{s} \right) - \frac{mgc}{RT} \right\}, \tag{1}$$

where $D$ is the effective diffusivity of a trace gas molecule ($m^2 \, s^{-1}$), the variables $s$, $c$, and $T$ are open porosity (unitless), trace gas molar concentration (mol $m^{-3}$), and firn temperature (K), respectively, and the constants $m$, $g$, and $R$ are the mass number of the trace gas (kg $mol^{-1}$), the acceleration of gravity (m $s^{-2}$), and the gas constant (J $mol^{-1}$ $K^{-1}$), respectively. Vertical advection flux of the trace gas, caused by air trapping at the close-off zone and downward bulk motion of firn, is expressed by using the equation given by Rommelaere et al. (1997). Conservation of the trace gas is given by

$$\frac{\partial c}{\partial t} + \frac{\partial (vc)}{\partial z} + \frac{\partial F}{\partial z} + rc = 0, \tag{2}$$

for the open pore space and

$$\frac{\partial c_b}{\partial t} + \frac{\partial (v_f c_b)}{\partial z} - rc = 0, \tag{3}$$

for bubbles. Here $c$ and $c_b$ are trace gas molar concentrations in the open pore space and bubbles, respectively. Vertical speed of air (m $yr^{-1}$) in the open pore space $v$ is distinguished from that of firn itself $v_f$. The vertical speed of firn $v_f(z)$ is simply given by dividing the accumulation rate by the firn density under the assumption of the steady-state densification of firn. At the transition zone where the open pore air is gradually trapped into bubbles, mass conservation is given by using a bubble trapping rate $r$ ($s^{-1}$), which simply means that a portion of the trace gas molar concentration in the open pore space ($rc$) is added to bubbles. The bubble trapping rate is given as a function of the open porosity, the total porosity, and the vertical speed of firn itself (Rommelaere et al., 1997). The total porosity was calculated from the firn density data. At the transition zone, the total porosity should be divided into the open and closed porosity. The closed porosity $s_c$ in the NGRIP firn was calculated by the empirical equation given by Schawander (1989):

$$s_c = s_t exp \left\{ 75 \left( \frac{\rho}{\rho_{close}} - 1 \right) \right\}, \tag{4}$$

where $s_t$ is the total porosity, and $\rho$ and $\rho_{close}$ are the density of firn and that at the close-off depth (kg $m^{-3}$), respectively. For the NEEM firn, the closed porosity $s_c$ was calculated as per Buizert et al. (2012) who followed the parameterization by Goujon et al. (2003):

$$s_c = 0.37 s_t \left( \frac{s_t}{s_{close}} \right)^{-7.6}, \tag{5}$$

where $s_{close}$ is the mean close-off porosity ($9.708 \times 10^{-2}$) according to Buizert et al. (2012).

## 3.2 Atmospheric scenarios

To simulate depth profiles of trace gases in firn, atmospheric histories of the target gases are required. In this study, we used atmospheric histories prepared by the NEEM firn-air modeling (Buizert et al., 2012) and by the CMIP6 experiments (Meinshausen et al., 2017) for all the trace gases presented in this study for the NGRIP and NEEM firn ($CH_4$, $CO_2$, $SF_6$, CFC-11, CFC-12, CFC-113, HFC-134a, $CH_3CCl_3$ and $^{14}CO_2$). Since Meinshausen et al. (2017) provides latitudinally gridded datasets, their historical data for the northernmost latitude (82.5° N) are used. Note that the $^{14}CO_2$ history for CMIP6 is available from another study (Graven et al., 2017), which is also used in this study. The $^{14}CO_2$ data by Graven et al. (2017) are available in $\Delta^{14}CO_2$ for three zonal bands (northern hemisphere, tropics and southern hemisphere), and were converted to $^{14}CO_2$ mole fraction as in Buizert et al. (2012). These atmospheric scenarios (hereafter referred to as the BZ and CMIP6 scenarios) are compared in Figure 2. The BZ (light blue) and CMIP6 (blue) Arctic scenarios show compatible historical trends in many trace gases. There are however slight differences in some trace gases between the two scenarios e.g. $SF_6$ and CFC-12, but we later show that these differences do not cause significant biases in reproducing their depth profiles at the NGRIP and NEEM firn sites.

In contrast, the Arctic $CH_4$ histories by the two studies differ considerably with maximum difference of ~85 ppb around 1910 (Figure 2b, see also Figure 1). The two $CH_4$ histories show similar trends after 1960, but before this, the data sets diverge further into the past. This disagreement is also clear in the interpolar difference (IPD), calculated relative to the CMIP6 histories for Antarctica (right axes). Note that the CMIP6 scenario for the Antarctic latitude was constructed based on the Law Dome ice core data (see agreement in Figure 1). While the BZ scenario shows a gradual increase in IPD, the CMIP6 scenario indicates almost constant values of IPD at ~130 ppb over the 20th century. This difference stems from the different methodologies that were used to produce the respective scenarios. As described by Buizert et al. (2012), the BZ $CH_4$ scenario was constructed by adding the presumed IPD to the Antarctic history (the Law Dome data), where the IPD was assumed to be proportionally correlated with the growth rate of $CH_4$. This seems a reasonable assumption, given that the IPD and growth rate are both largely subject to changes in emissions from the northern hemisphere (Dlugokencky et al., 2003; Ghosh et al., 2015; Chandra et al., 2021). Meinshausen et al. (2017) compiled historical measurement records from the worldwide networks as well as Antarctic/Greenland ice core and firn samples and constructed latitudinally gridded datasets of various greenhouse gases. For the historical trend of $CH_4$, they relied on the data set from the NEEM-S1 ice core by Rhodes et al. (2013) to produce the atmospheric histories for the northern hemisphere. They used the 5-yearly averaged values with outliers removed to represent lower bounds of the raw data points as shown in Figure 1. Note that the Rhodes et al. (2013) data set was not available when the BZ scenario was constructed. Thus $CH_4$ is unique because currently it has two diverging synthetic Arctic histories that are only loosely constrained by observational data.

This study assumes that the atmospheric scenarios for trace gases other than $CH_4$ are known with sufficient accuracy. The scenarios of the individual trace gases have inherent uncertainties, but the comparisons of the two available scenarios (BZ and CMIP6) indicate that the data sources for other gases do not show inconsistent variations as seen in $CH_4$. It should be however

noted that except CO₂, many trace gases lack observational data for the early 20th century, thereby both scenarios to a large extent being based on same data sources.

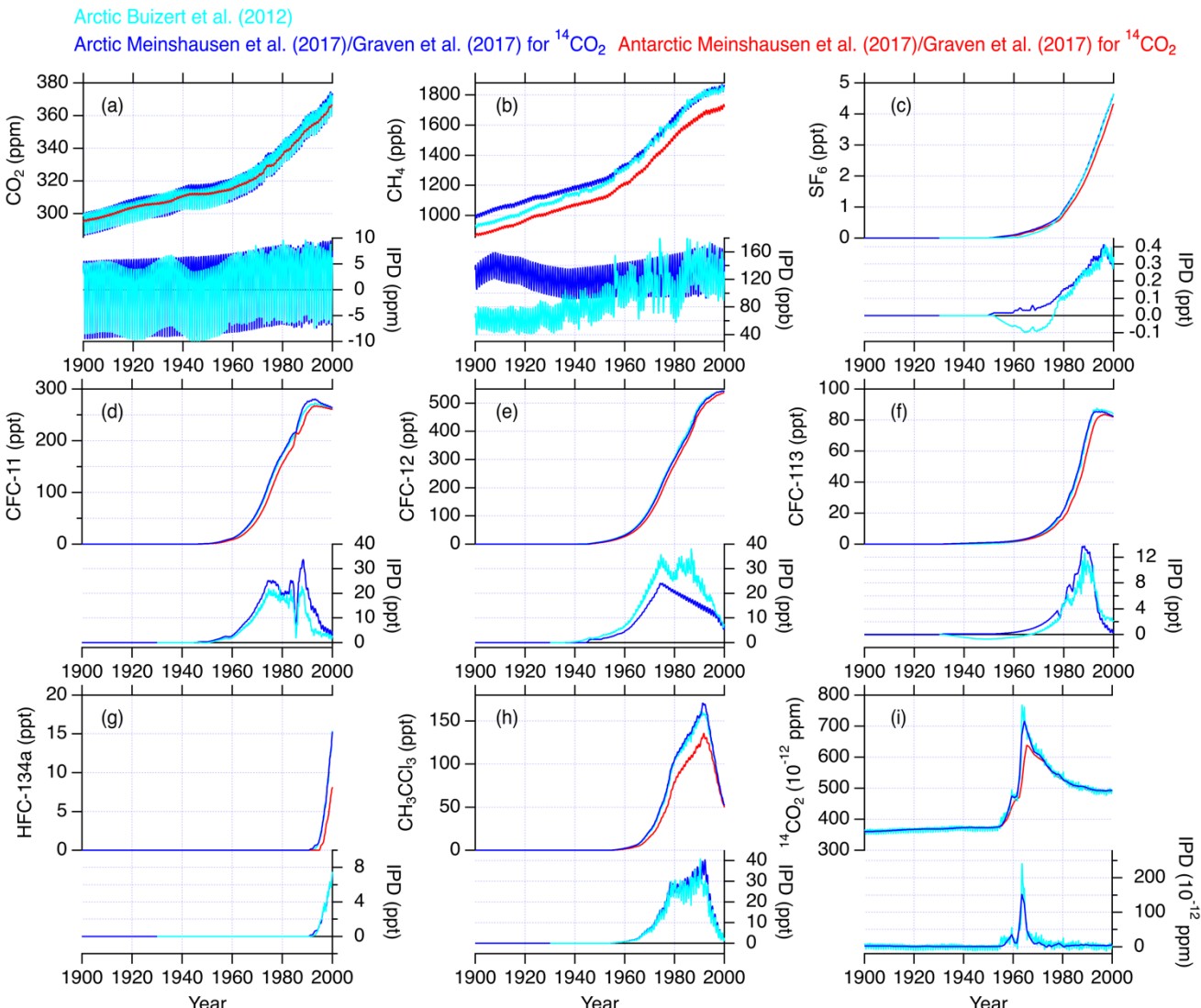

Figure 2: Atmospheric scenarios of various trace gases used in this study. The Arctic histories prepared for the NEEM firn-air
modeling (Buizert et al., 2012) are in light blue. The Arctic and Antarctic histories for CMIP6 (Meinshausen et al., 2017) are in blue and red, respectively. For ¹⁴CO₂, the CMIP6 historical data are from Graven et al. (2017). The interpolar differences (IPDs) calculated from the Arctic histories with respect to the Antarctic CMIP6 histories are also shown (right axes).

### 3.3 Effective diffusivity

We follow the previous firn-air studies in which the effective diffusivity in firn is optimized with an iterative method so as to minimize the difference between the simulated and observed depth profiles of $CO_2$ (Sugawara et al., 2003; Ishijima et al., 2007). In these previous studies, an initial guess of the depth profile of effective diffusivity for $CO_2$, $D_{init}(z)$, was calculated by:

$$D_{init}(z) = D_0 \left(\frac{T}{253}\right)^{1.85} \left(\frac{1013}{p}\right) \{1.7s(z) - 0.2\}, \tag{6}$$

where $s(z)$ and $D_0$ represent the open porosity at a depth $z$ and the diffusion coefficient of $CO_2$ at 253 K and 1013 hPa, respectively. $D_0$ was set to $1.247 \times 10^{-5}$ (m$^2$ s$^{-1}$) according to Trudinger et al. (1997). $p$ is the mean atmospheric pressure (Pa). The bulk density (kg m$^{-1}$) was determined by measuring the dimension and weight of cylindrically cut firn core samples (Kawamura et al., 2006). The effective diffusivity of $CO_2$ thus obtained, was converted to those of other trace gases by multiplying by scaling factors from Buizert et al. (2012). Therefore, the depth profile pattern of the effective diffusivity is

identical among all gases, but the magnitude is gas-dependent due to the scaling factors. In this study, the effective diffusivity profile prepared for the NGRIP firn by Ishijima et al. (2007) is referred to as the initial diffusivity and it was modified to improve the reproducibility of our newly measured trace gas profiles. For simulating trace gas profiles for the NEEM firn, we began with the effective diffusivity profiles available from Buizert et al. (2012). Those effective diffusivity profiles, which were originally optimised for individual firn-air transport models that participated in that study, were modified and used for

simulating the various trace gas profiles reported for the NEEM firn. The various diffusivity profiles were constructed by modifying the original profiles at a certain range of depths in a stepwise empirical manner; depth range of the diffusivity profile key to improve reproducibility of trace gas depth profiles was first diagnosed and then the diffusivity was perturbed up and down in the depth range to the degree in which the corresponding simulated profiles do not deviate substantially. Although this simple method does not guarantee identification of a best-match profile, we are confident that an acceptable range of the

diffusivity profile is satisfactorily constrained.

We eventually prepared 100 different sets of diffusivity profiles so as to cover a considerable range of diffusivity. Each set of the diffusivity profiles was evaluated based on the root mean square deviation (RMSD) between the model and data according to Buizert et al. (2012). All sets of effective diffusivity profiles for the NGRIP and NEEM firn sites are shown in Figure 3 (top and bottom panels, respectively). The different colors of the diffusivity profiles will be explained later. Those diffusivity

profiles were evaluated against the observed trace gas profiles, which were regarded as constraints. Note that $CH_4$ was not used in the evaluation and the atmospheric scenarios of other trace gases are assumed to be known with sufficient accuracy to infer a range of acceptable diffusivity profiles that reproduce the depth profiles of the firn-air composition.

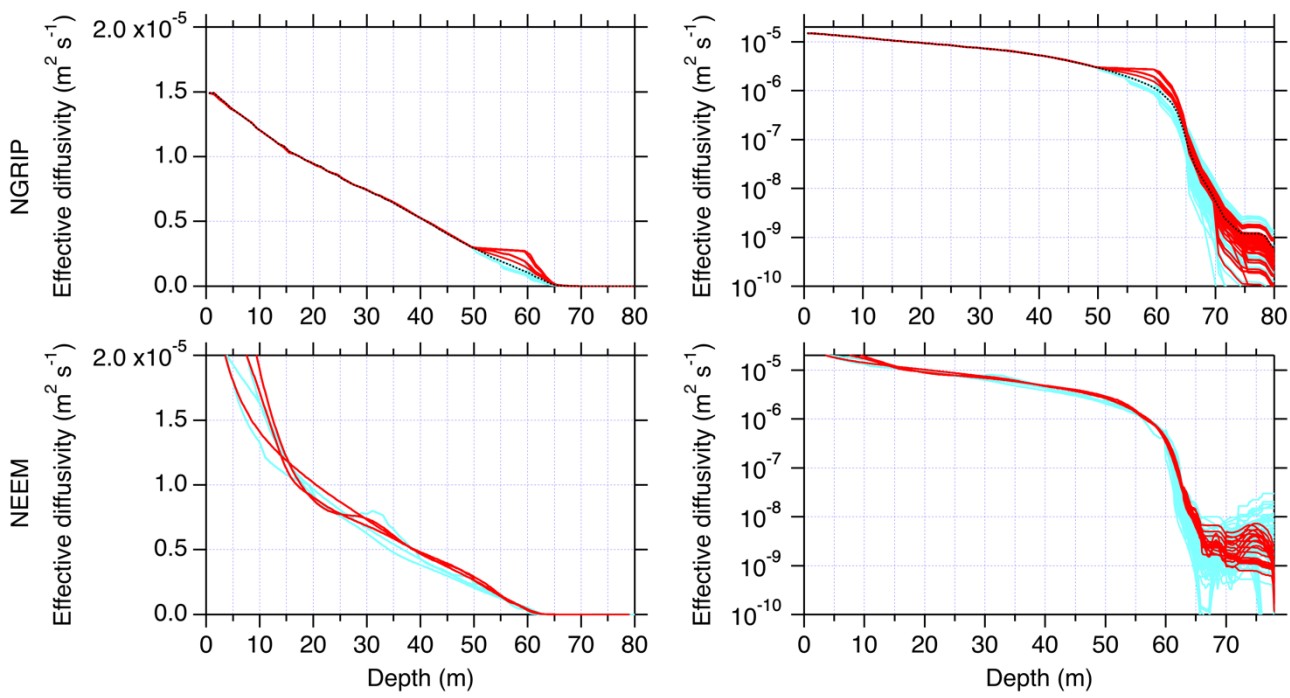

Figure 3: The 100 sets of effective diffusivity profile of $CO_2$ in the NGRIP (top panels, left panel on a linear scale and right panel on a log scale) and NEEM firn (bottom panels). The initial diffusivity profile (Ishijima et al., 2007) is shown by black dotted line (NGRIP only) and modified diffusivity profiles are in colors. The diffusivity profiles whose corresponding mole fraction profiles have RMSD values of <1.0 are colored red and the others light blue.

## 3.4 Performance of the firn-air transport model

To validate our firn-air transport model, we began by simulating depth profiles of various trace gases in the NEEM firn. Our model did not participate in the model intercomparison study using the NEEM data (Buizert et al., 2012). In this simulation, we employed the BZ scenarios as per their model intercomparison. The simulated depth profiles of the nine trace gases are presented in Figure 4 and compared with those by other models presented in Buizert et al. (2012). The results confirm that the performance of our model is comparable to those by other groups. As a measure of the model performance, Buizert et al. (2012) compared RMSD, which ranged from 0.73 to 0.92 for the six participating models. Following the same approach, our model yields the RMSD value of 0.83 for the NEEM EU borehole. This RMSD value was achieved with an effective diffusivity profile that was prepared by modifying the profile originally optimised for the CIC (Centre for Ice and Climate) model at a certain range of depths. Note that the RMSD value here was calculated including $CH_4$ as per Buizert et al. (2012), but, as described in section 3.5, $CH_4$ is excluded in calculation of RMSD in the following sections.

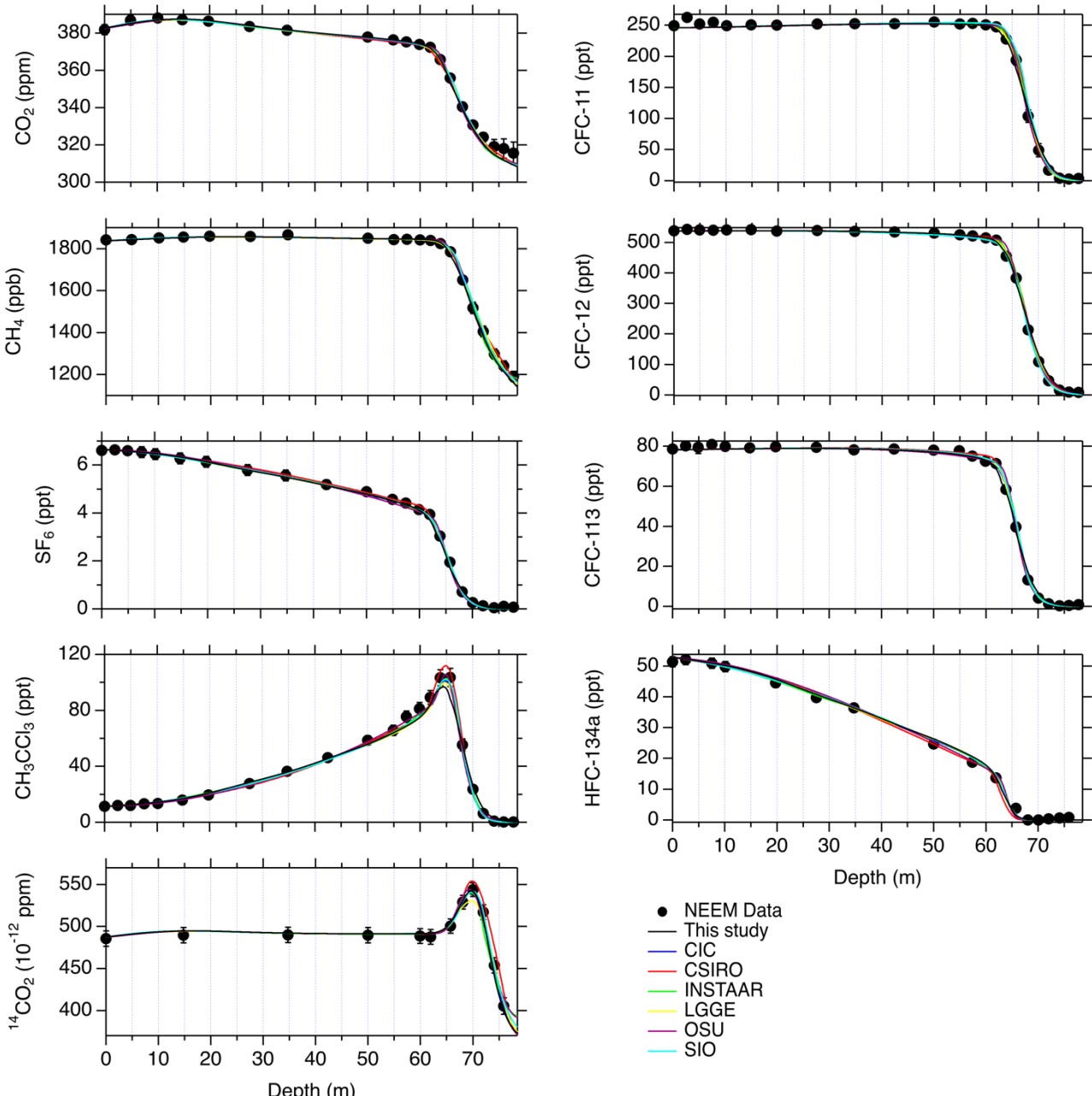


**Figure 4: Modeled depth profiles of various compounds in the NEEM firn. Closed circles in black are the observed data with uncertainty estimated by Buizert et al. (2012). Our model results are shown in black solid lines and results from other models are in colors. Different models are labeled by institutions according to Buizert et al. (2012): CIC (Centre for Ice and Climate), CSIRO (Commonwealth Scientific and Industrial Research Organization), INSTAAR (Institute of Arctic and Alpine Research), LGGE**

**(Laboratoire de Glaciologie et Géophysique de l'Environnement), OSU (Oregon State University) and SIO (Scripps Institution of Oceanography).**

### 3.5 Reconstruction of CH$_4$ history

Our modeling procedure for reconstructing the atmospheric histories of CH$_4$ was as follows:

(1) To represent atmospheric trends of different trace gases in the Arctic region, we began by employing the two sets of atmospheric scenarios (section 3.2). The firn transport model calculates depth profiles of the various trace gases at the NGRIP and NEEM firn sites using the large set of modified ($N$=100) effective diffusivities described in section 3.3. The simulation case with each diffusivity profile was evaluated based on RMSD. As we aim to estimate the historical atmospheric trend of CH$_4$, the RMSD-based evaluations were made using all of the available trace gas data, excluding CH$_4$. In the RMSD calculation, we used the measurement uncertainties of 0.2 ppm for CO$_2$, 0.2 ppt for SF$_6$, 1.1 ppt for CFC-11, 3.3 ppt for CFC-12, 0.6 ppt for CFC-113 and 3.2 ppt for CH$_3$CCl$_3$ for the NGRIP firn. For the NEEM firn, we employed the uncertainties provided by Buizert et al. (2012). It should be noted that the present study does not follow the uncertainty estimation as done by Buizert et al. (2012). They indicated that uncertainties in the atmospheric scenarios as well as measurement uncertainties are the two largest contributors to the total uncertainties for individual data points for the NEEM firn. We consider that the uncertainties in the atmospheric scenarios are appreciably examined through comparisons of series of simulations using the two independent scenarios.

(2) We ran the model with 100 different sets of diffusivity profiles to calculate the depth profile of CH$_4$. Note that, based on the earlier step, we know the diffusivity profiles that generate reasonable firn-air profiles for the trace gases other than CH$_4$. Every diffusivity profile was used in combination with the firn-air CH$_4$ data for reconstructing an atmospheric CH$_4$ history. We employed an iterative dating approach (Trudinger et al., 2002) where the initial atmospheric scenario (the BZ scenario) was modified to improve model reproducibility of the CH$_4$ depth profile (see below). The corrected atmospheric CH$_4$ scenarios were then compared to the original scenarios (BZ and CMIP6) for further discussion.

The iterative dating for CH$_4$ was performed as follows:

(*I*) Depth profile of CH$_4$ was calculated with the initial atmospheric CH$_4$ scenario.

(*II*) The modeled CH$_4$ mole fraction, calculated in step *I*, was compared to the input atmospheric CH$_4$ scenario, and effective age at each sampling depth was determined as the time when the modeled CH$_4$ agreed with a value in the atmospheric CH$_4$ scenario. It is noted that the smoothing spline curve applied to the BZ CH$_4$ scenario was used for calculation of the effective age, as the input scenario with seasonal variation (Figure 2) would not allow the effective age to be uniquely determined.

(*III*) A new atmospheric CH$_4$ scenario was constructed by assigning the observed CH$_4$ mole fraction, at each sample depth, to the effective age determined in step *II*. The observed CH$_4$ versus the effective age data set was interpolated by a smoothing spline function and it is considered as a revised atmospheric CH$_4$ scenario.

(*IX*) Depth profile of CH$_4$ was again calculated with the revised atmospheric CH$_4$ scenario constructed in step *III*.

(*X*) The above steps *II*–*IX* were repeated until the model-data difference converged within an acceptable range (typically after a few iterations) (Trudinger et al., 2002; Ishijima et al., 2007). In this study, we made five iterations for each modified diffusivity case as we confirmed sufficient convergence of the result.

## 4 Result

### 4.1 Initial simulations: NGRIP

Figure 5 presents the simulation results with the initial diffusivity in comparison to the observed profiles for the six trace gases excluding $CH_4$ ($CO_2$, $SF_6$, CFC-11, CFC-12, CFC-113 and $CH_3CCl_3$) for the NGRIP firn. It is again noted that the initial diffusivity profile was tuned only for the depth profile of $CO_2$ (Ishijima et al., 2007). As seen in this figure, measured profiles of these trace gases (except $CH_3CCl_3$) show gradual decreases with depth in the DZ and sharp decreases in the LIZ. In contrast, $CH_3CCl_3$ increases with depth in the DZ and sharply decreases in the LIZ. The difference of the depth profile pattern among species is due to their different historical atmospheric trends. It is known that, since the mid 20th century, the atmospheric mole fractions of the five trace gases ($CO_2$, $SF_6$, CFC-11, CFC-12 and CFC-113) have increased either monotonically or shown peak/slowed increase in the early 1990s (Sturrock et al., 2002; Martinerie et al., 2009). In contrast, $CH_3CCl_3$ has increased until the early 1990s and has rapidly decreased since then (Sturrock et al., 2002; Rigby et al., 2017), which is also observed in Figure 2. Our simulation reproduces the observed depth profiles of these six trace gases in the NGRIP firn fairly accurately using the BZ scenarios.

335

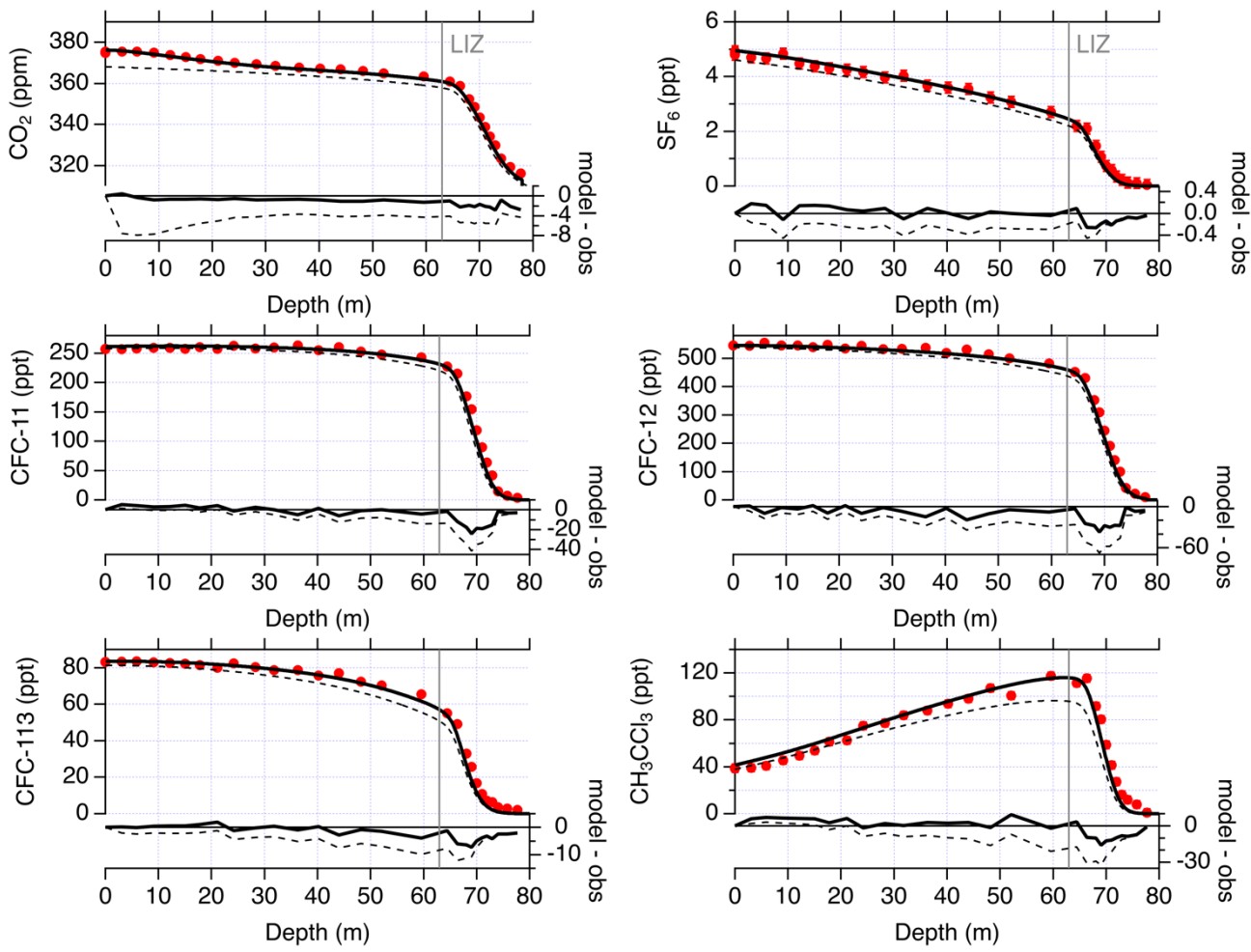

**Figure 5: Depth profiles of CO₂, SF₆, CFC-11, CFC-12, CFC-113 and CH₃CCl₃ in the NGRIP firn. The measurement and model results are shown by red circles and black solid lines, respectively (left axis). The measurement uncertainties are shown as vertical error bars though in many cases they are smaller than the circle sizes. Black solid lines show the modeled profiles with the initial diffusivity and the BZ scenarios. The black dashed lines indicate the profiles calculated with the atmospheric scenarios for Antarctica**
340 **(red lines in Figure 2). The model-data differences are also shown (right axis). The vertical solid line in each panel indicates the upper depth of LIZ.**

It is interesting to note that the depth profiles of $CH_3CCl_3$ at NGRIP and NEEM are remarkably different. Whereas the NEEM data show a relatively sharp $CH_3CCl_3$ peak in the LIZ (~65 m, Figure 4), NGRIP does not show such a narrow peak. This is

345 due to the timing of firn-air sampling i.e. 2001 for NGRIP and 2008 for NEEM. When the NGRIP firn air was sampled, the signal of the maximum atmospheric $CH_3CCl_3$ in the early 1990s had only reached near the top of the LIZ at the site, thereby formulating the relatively gentle changes at the shallower depths. On the other hand, seven years later at the NEEM site, such a signal was found deeper in the LIZ where the age of air changes rapidly with depth in both deeper and shallower sides. We

emphasize that, despite the differences in the depth profiles of $CH_3CCl_3$ at the two sites, our simulations reproduce the profiles measured at both sites well, using the same atmospheric $CH_3CCl_3$ scenario.

## 4.2 Sensitivity to diffusivity profile: NGRIP

Figure 5 shows that the model-data difference increases in the LIZ for all the trace gases. In particular, the model-data difference is pronounced as a dip around 70 m for all trace gases, implying that the mismatches may originate in a common factor in the modeling e.g. depth profile of diffusivity. To examine the impact of diffusivity modification on the simulated depth profiles and their agreements with the data, we examined the 100 sets of modified diffusivity profiles (Figure 3). It was found that, to reduce the model-data difference in the LIZ (Figure 5), the diffusivity needs to be increased in the shallower layers compared with the top LIZ i.e. 50–65 m. The diffusivity was also modified in the deeper layers (>65 m) and simulations were made accordingly. The simulated profiles for depths deeper than 50 m using the 100 diffusivity profiles are presented for the six trace gases (Figure 6). In this figure, the modeling results with the modified diffusivity profiles are shown in colors on the left axis, and the model-data differences of the respective cases are plotted on the right axis. It is clear that the model-data differences could be significantly reduced with some diffusivity cases. The RMSD values are as little as 0.51 for a particular case. In Figure 6 and associated figures, the model results with RMSD of <1.0 are colored red and other cases are colored light blue.

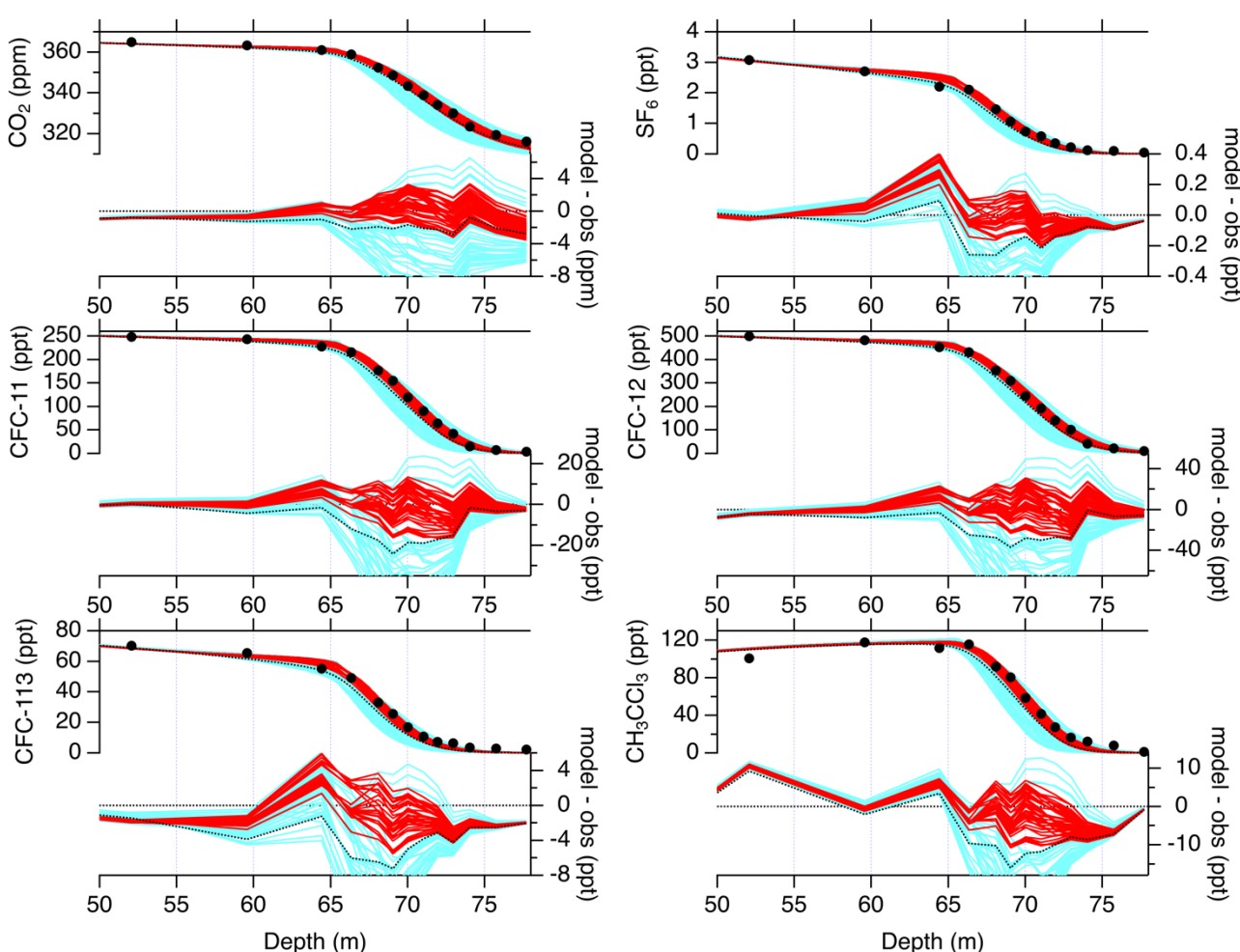

Figure 6: Depth profiles of the six trace gases below 50 m depth in the NGRIP firn. Black circles indicate the measurements and solid lines in colors are the model results with different diffusivity profiles and the BZ scenarios (left axis). Black dotted lines are modeled results with the initial diffusivity. Also shown are the model-data differences (right axis). See text for difference in line colors.

## 4.3 Sensitivity to atmospheric scenario: NGRIP

In Figure 7, modeled depth profiles with different sets of the atmospheric scenarios (BZ and CMIP6) are compared for trace gases other than $CH_4$. For simplicity, only the results with RMSD of <1.0 are presented. This figure shows that the differences in the atmospheric histories (Figure 2) produces relatively small differences in the depth profiles in the firn. There are small offsets due to the differences of the histories in some gases; difference in the $SF_6$ history before 1980 (<0.2 ppt) corresponds to the small (<0.1 ppt) offsets below 65 m; significant differences in the histories of CFC-11 (<10 ppt) and CFC-12 (<20 ppt) for 1960–1990 resulted in the overall offsets (roughly <5 and <10 ppt, respectively) below 50 m;

difference in the $CH_3CCl_3$ history (<10 ppt) in the early 1990s produced the offsets (~3 ppt) above 66 m. The smaller offsets in the calculated depth profiles than in the input atmospheric scenarios are due to the smoothing effect of diffusion in the firn layers. We calculated the difference between the modeled profiles with the two scenarios for individual diffusivity cases and

found that those differences in the LIZ are within the measurement precisions for $SF_6$, CFC-113 and $CH_3CCl_3$. They are a bit larger for $CO_2$, CFC-11 and CFC-12, which are up to five times the respective measurement precisions.

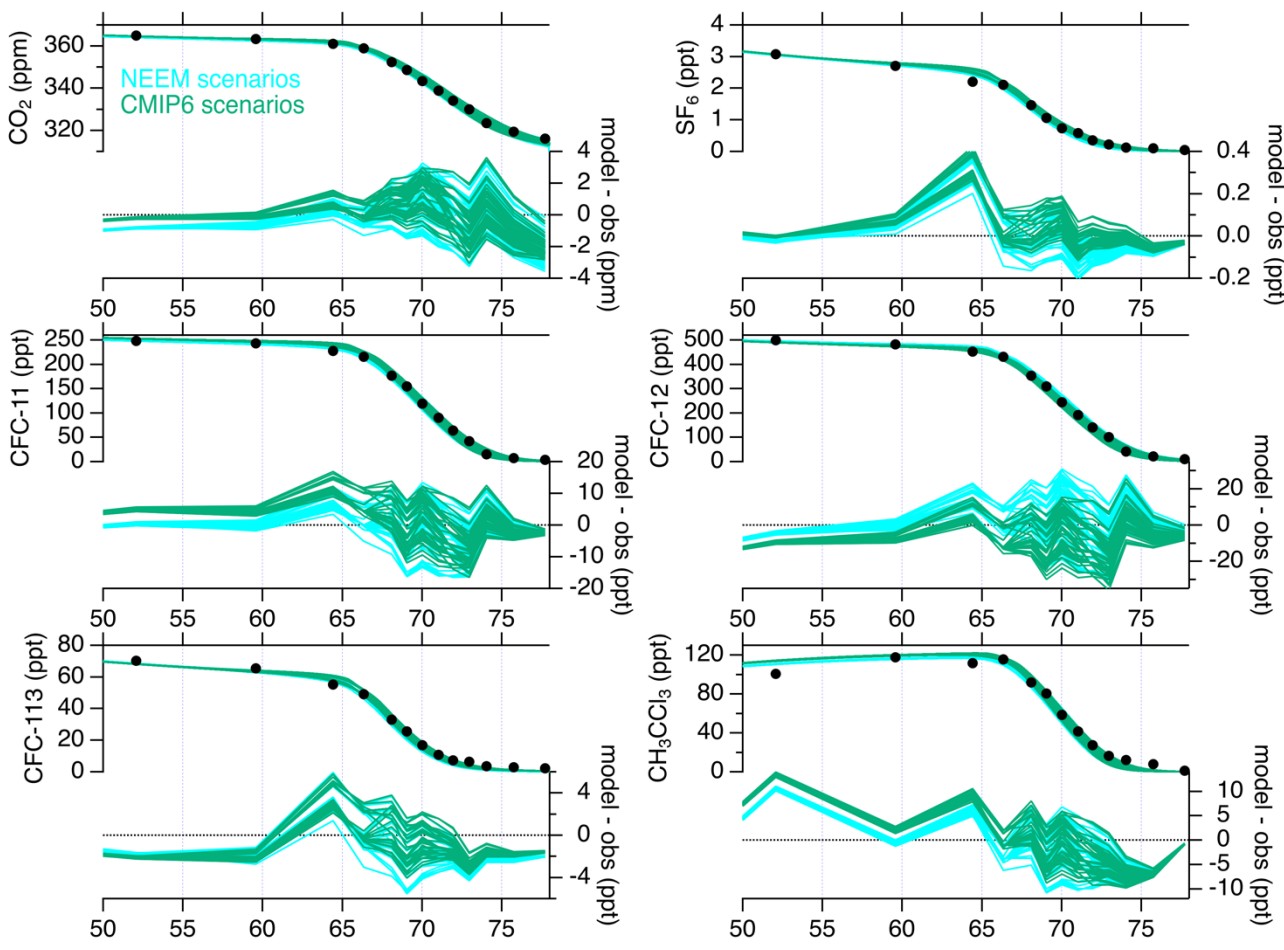

**Figure 7: Same as Figure 6, but for comparison of the modeled profiles in the NGRIP firn with different atmospheric scenarios of BZ and CMIP6 in light blue and green, respectively. Only the simulation results with RMSD values of <1.0 are shown.**


## 4.4 Simulations of $CH_4$: NGRIP

Modeled depth profiles of $CH_4$ are shown in Figure 8. These calculations were made with the 100 diffusivity profiles and the atmospheric $CH_4$ scenarios of BZ (left) and CMIP6 (right). It is interesting to note that the characteristics of the model-data

difference for CH$_4$ are different from those for the other six trace gases (Figure 6). For the other trace gases, the initial

simulation showed increased model-data differences around 70 m, and they were reduced with some modified diffusivity profiles. For CH$_4$, the model run with the initial diffusivity profile and the BZ scenario reproduces the observed CH$_4$ profile quite well down to ~73 m, but significantly overestimates by >20 ppb for the lowest three depths (the black dotted line in the left panel). Using the modified diffusivity profiles that allowed better agreements for the other six trace gases (red lines), we find larger model-data CH$_4$ differences than in the initial simulation. These features are also seen for the simulation results

with the CMIP6 scenario (right panel), but the overestimate in the LIZ is more pronounced, because the CH$_4$ mole fractions for the early to mid-20th century are higher in the CMIP6 than in the BZ scenario. In comparison to other gases (section 4.3), the largest impact of the scenario difference occurs in CH$_4$. In the LIZ, the difference between depth profiles with the two scenarios reaches to 5–10 times the measurement precision, namely >30 ppb in some diffusivity cases.

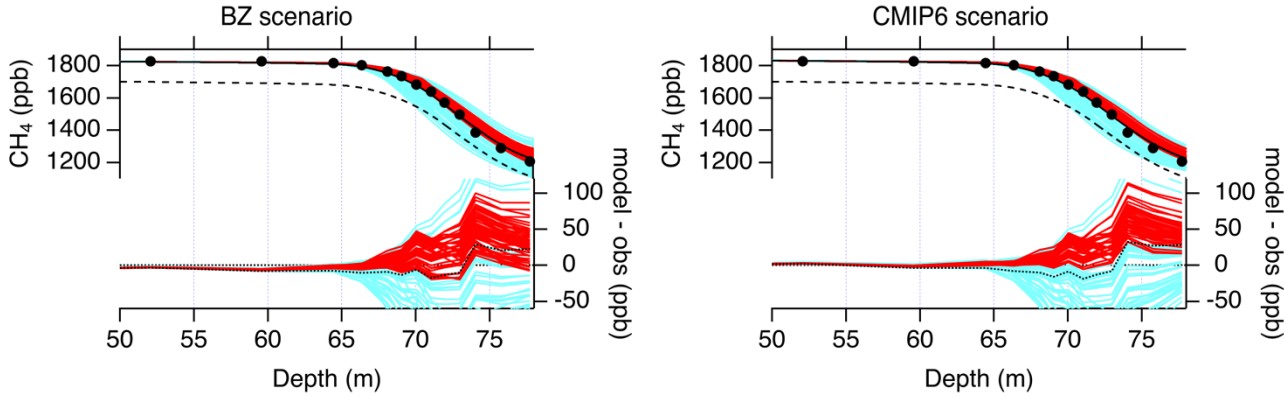


**Figure 8: Same as Figure 6, but for CH$_4$ in the NGRIP firn. Simulation results with the atmospheric scenarios of BZ and CMIP6 are shown in left and right panels, respectively. The black dashed lines indicate the profiles calculated with the atmospheric scenarios for Antarctica (red line in Figure 2).**

**4.5 Sensitivity to IPD: NGRIP**

Reflecting geographical source/sink allocation and atmospheric lifetime, different trace gases show different magnitude of IPD as shown in Figure 2. It would be possible for some gases that northern to southern difference in mole fraction have little influence on depth profile in firn due to relatively small interhemispheric gradient. To examine magnitude of the sensitivity of the depth profiles to the IPD for individual trace gases, we calculated depth profiles that would be expected if the Antarctic

atmospheric scenarios (red lines in Figure 2) are given to force the firn model for the NGRIP firn. The results are shown by dashed lines in Figure 5. This sensitivity experiment shows that IPD causes significant biases larger than the measurement precisions of the respective trace gases. We calculated the difference between the simulations for the BZ scenario (solid line) and the Antarctic scenario (dashed line) and found that sensitivities to the IPD for these six trace gases are no more than 20

times the respective measurement uncertainties. Such relative sensitivities of these suite of gases to the IPDs are much smaller
than that of CH₄, which reaches 40 times the measurement uncertainty. As seen in Figure 8, the calculated CH₄ profile with
the Antarctic scenario for the NGRIP firn is aligned ~120 ppb below the original simulation, showing the pronounced impact
on CH₄.

## 4.6 Sensitivity to atmospheric scenario: NEEM

In addition to the above simulations for the NGRIP firn, we examined the impact of the difference of the atmospheric scenarios
on the depth profiles for the NEEM firn. In Figure 9, comparisons between simulations with the different scenarios for the
nine trace gases including CH₄ are presented. We found some common characteristics at the NGRIP and NEEM firn sites.
While the difference between the simulations with the two scenarios are relatively small for most trace gases, the large
difference between the CH₄ scenarios (i.e. higher CH₄ mole fraction in the CMIP6 scenario, see Figure 2) results in
overestimation of CH₄ mole fraction in the LIZ (> 63 m) in the modeled profiles with the CMIP6 scenario. In the LIZ, the
difference of the depth profiles with the two scenario exceeds 30 ppb, being comparable to the magnitude observed in the
NGRIP firn (Figure 8).

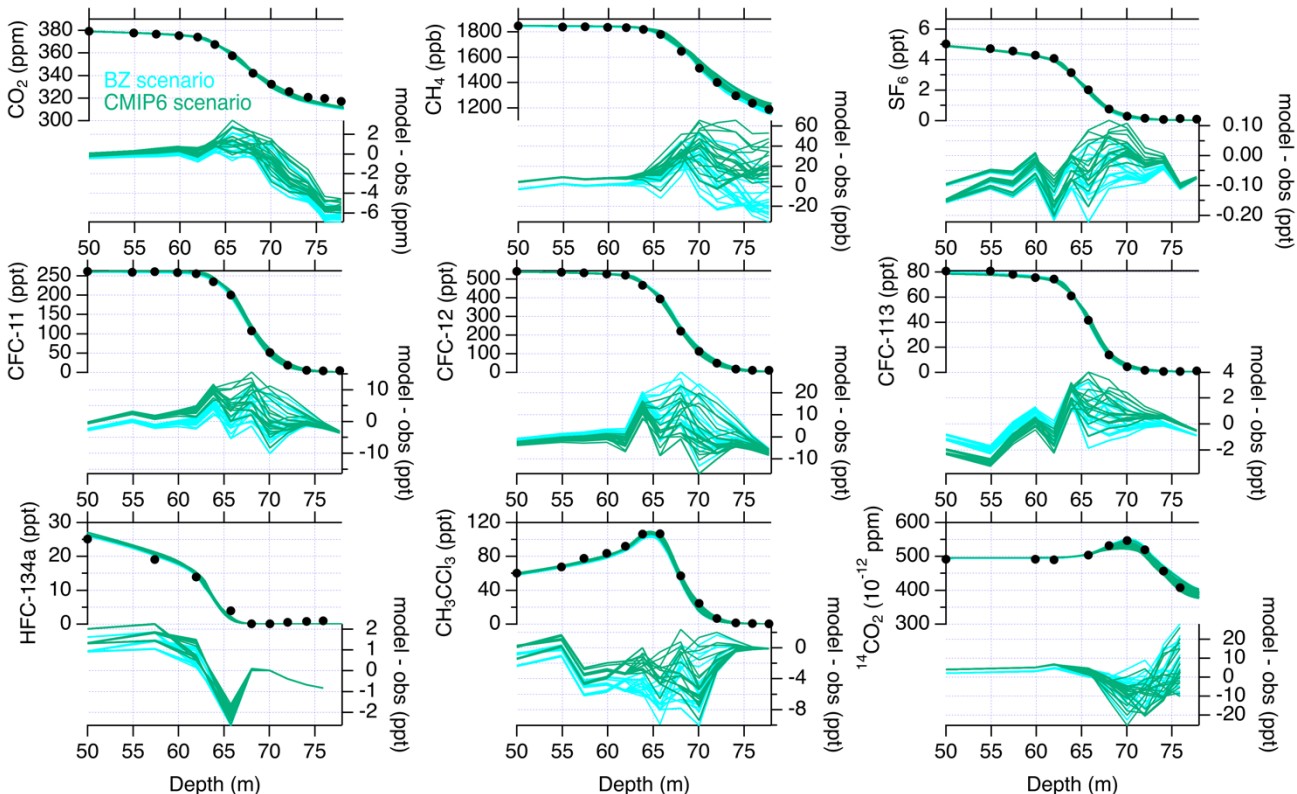

**Figure 9. Same as Figure 7, but for the NEEM firn. Only the simulation results with RMSD values of <1.0 are shown.**

### 4.7 Reconstruction by iterative dating

As shown in the earlier sections, $CH_4$ shows uniquely high uncertainty in the firn modeling compared to other trace gases. First, simulated depth profiles of $CH_4$ do not show improved reproducibility with effective diffusivity profiles that satisfy depth profiles of other gases (sections 4.2 and 4.4). Second, while other gases show limited sensitivities to the different atmospheric scenarios, inconsistency between the currently available two scenarios have particularly large impact on the simulated depth profile of $CH_4$ (sections 4.3 and 4.4). Third, sensitivity to IPD in the atmospheric scenarios is most pronounced for $CH_4$ compared to other trace gases (section 4.5). These characteristics are common for both NGRIP and NEEM sites (section 4.6).

Taking all these unique characteristics of $CH_4$ into account, we explore the possibility of reconstructing the Arctic $CH_4$ mole fractions by the iterative dating method. This approach has a fixed diffusivity profile (assumed to be correct) and aims to find an acceptable atmospheric history to reproduce the firn-air depth profile. It is considered that the modified diffusivity profiles with the low RMSD values are adequately evaluated by the trace gases except $CH_4$ and that the model-data mismatch in the $CH_4$ modeling is therefore attributable to uncertainty in the atmospheric $CH_4$ scenario. The historical atmospheric $CH_4$ variations obtained by the iterative dating method are presented in Figure 10 for the 100 modeling cases of the modified diffusivity. The different simulation cases are colored in the same manner as in the earlier figures according to the RMSD (Figures 3, 6 and 8). Note that the reconstruction cases colored in light blue are considered to be less likely, due to poorer reproduction of the depth profiles (Figures 6 and 8). The NGRIP reconstruction results (red, Figure 10a) are in good agreement with the BZ scenario after around 1980. For the earlier period, however, the upper bounds of the reconstructions (line connecting far-left red circles) match with the BZ scenario, and the overall range of acceptable histories is below the BZ scenario (circles and shades in red). On the other hand, the NEEM reconstruction results show that the range of acceptable histories are distributed closely around the original BZ scenario or below it after around 1960, whereas the lower bounds of the reconstruction (line connecting far-right red circles) aligns with the BZ scenario before that (Figure 10b).

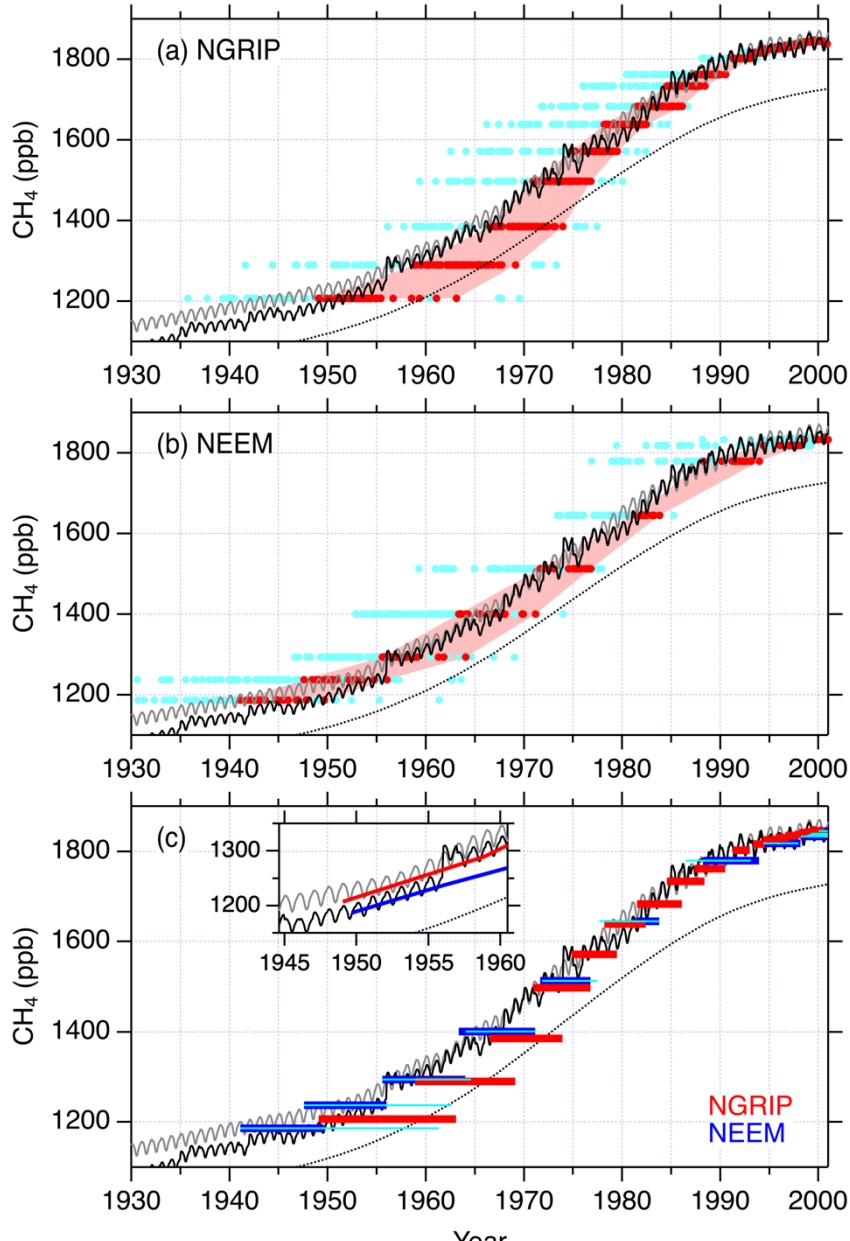

Figure 10: Results of reconstructions by the iterative dating method for the model cases with the 100 different diffusivity profiles (circles colored in the same manner as in Figures 3, 6 and 8) from the (a) NGRIP and (b) NEEM firn air, and (c) the ranges of reconstructions from both firn. The red shades indicate the range of CH₄ mole fraction trends reconstructed using the diffusivity profiles that better reproduce the trace gases except CH₄. Black and grey solid lines are the atmospheric CH₄ scenarios of BZ and CMIP6, respectively, and black dotted line is the smoothed Antarctic CH₄ mole fraction (spline curve) from the Antarctic data shown in Figure 1. The thick horizontal bars in red and blue in panel c correspond to red shades in panels a and b. The light blue thin bars in panel c indicate the range of the reconstructions for cases in which ¹⁴CO₂ data are excluded for evaluations of the diffusivity profiles. The inset in panel c shows zoom-in view of the NGRIP upper bounds (red) and the NEEM lower bounds (blue) in comparison to the two scenarios.

It should be stressed that the reconstructions for the time period before 1980 rely on $CH_4$ data from the lowest four or five depths of the NGRIP and NEEM firn air. The differences between the initial and corrected atmospheric $CH_4$ scenario from the three deepest data for the NGRIP firn are up to ~100 ppb. As seen in Figure 10, such reduction in the Arctic atmospheric $CH_4$ scenario over the period would result in alignment with the atmospheric $CH_4$ trend in Antarctica inferred from the Law Dome and other datasets (black dotted line). It is again noted that all the reconstruction cases colored in red used diffusivity profiles that yield relatively good model reproducibility with RMSD values of <1.0. It is therefore seen that iterative dating-based reconstruction from the NGRIP firn data suggests lower $CH_4$ mole fraction than the BZ or CMIP6 scenarios from the 1950s to 1970s in any case, albeit with large uncertainty. It is however noted that, as shown in Figure 10c, this result is not fully compatible with that from the NEEM firn. In particular, the discrepancies between the reconstructions from both firn data diverge with time before 1970. When compared to these reconstructions, the BZ scenario (black line) tracks the overlapping ranges of both reconstructions, while the CMIP6 scenario passes above them before around 1955.

## 5 Discussion and conclusion

### 5.1 $CH_4$ history

In section 4, we have shown that, with the two atmospheric scenarios (BZ and CMIP6), depth profiles of $CO_2$, $SF_6$, CFC-11, CFC-12, CFC-113 and $CH_3CCl_3$ in the NGRIP and NEEM firn are reproduced with sufficient accuracy by using range of modified diffusivity profiles (Figures 6, 7 and 9). This suggests that the atmospheric scenarios of these trace gases used in the modeling are consistent with the depth profiles observed at both firn sites in Greenland (NEEM and NGRIP). In contrast, the observed $CH_4$ profile in the NGRIP firn was not accurately reconciled using the two atmospheric scenarios and the diffusivity profiles that allow adequate reproducibility for the trace gases (except $CH_4$). This suggests either that the Arctic atmospheric scenarios of $CH_4$ are uncertain or that the diffusivity profile of the NGRIP firn is underconstrained. We explored the correction of the atmospheric scenario of $CH_4$ by the iterative dating approach. This method improves agreement to the observed $CH_4$ depth profile, with an implicit assumption that the diffusivity profile in each case is correct. Although uncertainty due to the under-constrained diffusivity profile in the LIZ is large, this attempt for the NGRIP firn suggested that the $CH_4$ mole fractions over the period 1950–1980 could be ~100 ppb lower than the original BZ scenario (Figure 10). In contrast, the iterative dating reconstruction for the NEEM firn agrees with the BZ scenario. Although the spread of the reconstructions is large, particularly for the NGRIP firn, it was found that the BZ scenario passes within the ranges of the reconstructions from both firn data, but that the CMIP6 scenario is notably higher than the reconstructions for the period before 1960.

Whereas uncertainties in the reconstructions from the individual firn sites are large, Figure 11c could suggest a relatively narrow range of the $CH_4$ history that satisfies both NGRIP and NEEM reconstructions. Although the overlapping range of the two reconstruction is ~90 ppb in the 1970s, it is as small as ~30 ppb in the 1950s. This suggests that the combined NGRIP and

NEEM firn data could provide a stronger constraint to the range of the $CH_4$ mole fraction e.g. 1185–1215 and 1225–1260 ppb in 1950 and 1955, respectively (Figure 10c inset). It is again noted that only the BZ scenario fall within these ranges for the period, suggesting that it is likely closer to the true atmospheric $CH_4$ history than the CMIP6 scenario for the mid 20th century. The IPD of the atmospheric $CH_4$ mole fraction is important for better understanding the evolution of the global $CH_4$ budget.

Given that the Antarctic ice core and firn measurements have provided relatively reliable $CH_4$ records over the 20th century, improved reconstructions from Greenland ice cores and firn air should better constrain the changes in the IPD. To sufficiently constrain the historical global $CH_4$ budget, the reconstruction for Greenland needs to be accurate within ~10 ppb, corresponding to ~30 Tg $CH_4$ $yr^{-1}$ global emission. Based on the currently best firn $CH_4$ data from NGRIP and NEEM, we demonstrated that consistent reconstruction of the Arctic $CH_4$ mole fraction is achievable back to the 1950s, but the uncertainty of reconstruction

is still large (> 30 ppb) for the 1950s to 1970s.

## 5.2 Uncertainty of effective age

It is important to note that the reconstructions for the period before 1980 from the NGRIP firn were heaviliy influenced by the five deepest data in the LIZ (> 72 m). Figure 11a shows distributions of the effective age of $CH_4$ at depths below 55 m in the NGRIP firn, colored the same as in the earlier figures. In addition, the spread of the effective age ($\sigma_{age}$) at each sampling depth

is shown on the right axis. This figure shows that firn-air samples collected at the five lowest sampling depths at NGRIP have effective ages corresponding to the period from ~1950 to the late 1970s. At those depths, even the acceptable diffusivities yield the spread of the effective age of >5 years (black vertical bars). This shows that the reconstruction of the $CH_4$ mole fraction for the period is subject to much uncertainty in effective age. For the NEEM firn, the reconstructions before 1980 also rely on the five deepest data in the LIZ (Figure 11b). The $\sigma_{age}$ values at those depths in the NEEM firn ranges from 5 to 8 years (thick

vertical bars), comparable to those in the NGRIP firn. The Antarctic atmospheric $CH_4$ record (see Figures 1 and 10) indicates that the atmospheric increase rate of $CH_4$ was 10–15 ppb $yr^{-1}$ over the period. The 5-year uncertainty in the age estimate for the NGRIP firn-air samples could therefore be translated to an uncertainty of >50 ppb in the Arctic atmospheric $CH_4$ level. This is comparable to the IPD of $CH_4$ mole fraction during the 1950s to 1970s, which was assumed when Buizert et al. (2012) prepared the BZ $CH_4$ scenario.

It is also interesting to note that $\sigma_{age}$ at the four deepest depths in the NEEM firn is almost constant, whereas it increases with depth in the NGRIP firn and exceeds 10 years at the two deepest depths. This indicates that effective age in the oldest firn-air layers can be estimated with better accuracy in the NEEM firn than in the NGRIP firn, thereby providing the reconstructions with smaller uncertainties. As $\sigma_{age}$ was calculated from the simulation cases with acceptable ranges of diffusivity, its magnitude reflects how tightly the diffusivity profile is constrained at each firn site. In other words, our simulations infer that the

diffusivity profile in the NEEM firn can be better constrained than the NGRIP firn.

## 5.3 Constraint to diffusivity profile

In order to see the degree of constraint to the effective diffusivity from different trace gases, we calculated RMSD values for different combinations of the trace gases for the NEEM firn. The choice of the NEEM firn is due to the availability of a larger number of gas species. It was found that the $^{14}CO_2$ data provide strong constraints for narrowing the acceptable range of

diffusivity profiles in the LIZ as its history has a unique peak in the early 1960s due to nuclear weapons testing (Figure 2). The evaluated RMSD with the $^{14}CO_2$ data excluded and the historical $CH_4$ reconstruction from the simulation cases with RMSD <1.0 is presented in Figure 10c (thin horizontal bars in light blue). The corresponding spread of the effective age is also shown in Figure 11b (thin vertical bars in black). These results show that, for the NEEM reconstruction, uncertainty of the effective age would be doubled at the two deepest depths if it were not for the $^{14}CO_2$ data. It is also interesting that the range of the $CH_4$

reconstruction without $^{14}CO_2$ would deviate to the younger ages and would suggest a historical trend lower than the BZ scenario (Figure 10c). In turn, the $CH_4$ trend reconstructed from the NGRIP firn might be different if $^{14}CO_2$ measurement was available. The above contribution of the $^{14}CO_2$ data for the NEEM firn implies that the NGRIP reconstruction could have been closer to the NEEM reconstruction. It is also noted that constraints from halocarbon species to the diffusivity profile are relatively weak as their mole fractions in the LIZ are decreased to close to zero.

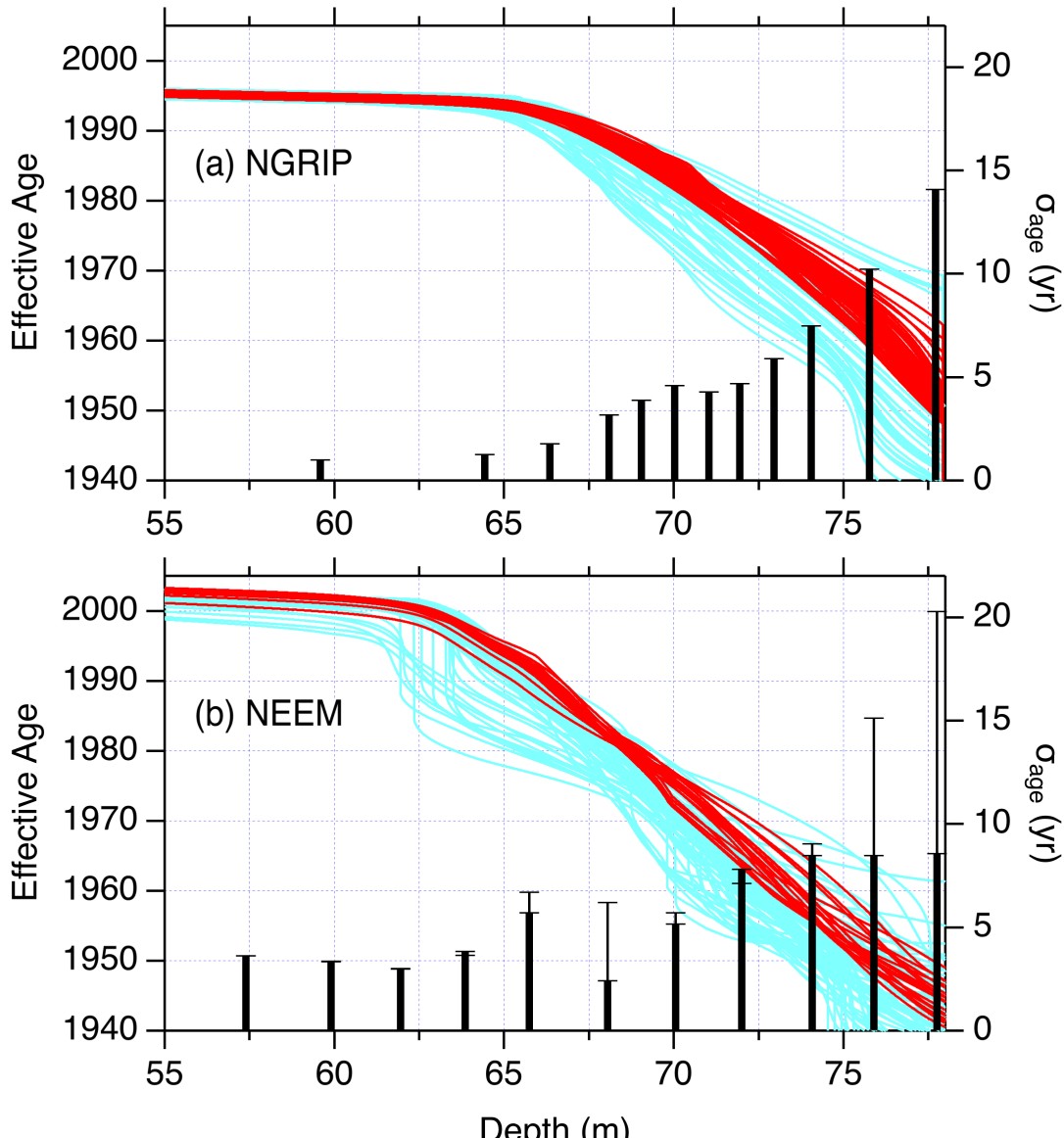


**Figure 11: Depth profiles of effective age of CH₄ calculated after the iterative dating calculations for depths below 55 m at the (a) NGRIP and (b) NEEM firn sites (left axis). The solid lines represent cases for different diffusivity profiles and are colored in the same manner as in the earlier figures. Vertical bars in black indicate spread of the effective age ($\sigma_{age}$, maximum minus minimum) at the individual sampling depths among the modeling cases in red (right axis). For the NEEM firn, the thin vertical bars correspond**

**to the spread for simulation cases in which $^{14}CO_2$ data are excluded for evaluations of the diffusivity profiles.**

## 5.4 Outlook

Previous studies have used $CH_4$ as one of tracers that constrain diffusivity profiles in firn (Witrant et al., 2012; Trudinger et al., 2013). These studies have shown that $CH_4$ effectively contributed to constraining diffusivity for deep layers i.e. LIZ in the NEEM firn. However, the use of $CH_4$ as an effective constraint is valid only when its atmospheric scenario given as input to the models was assumed to be correct. Figures 6 and 8 show a larger spread in the model-data differences of $CH_4$ among the different diffusivity cases spread widely in the deepest layers of the LIZ of the NGRIP firn, in comparison to the other six gases. This fact highlights that subtle changes of the diffusivity profile in the LIZ have a large impact on simulating $CH_4$, and it indeed indicates that the species could serve as an effective diffusivity constraint, if its atmospheric scenario was correctly given with low uncertainty. This study indicated that the two currently available Greenland firn data sets (NGRIP and NEEM) are more consistent with the BZ $CH_4$ scenario (Buizert et al., 2012) than the CMIP6 scenario (Meinshausen et al., 2017). Furthermore, it should be again pointed out that the CMIP6 scenario suggests an almost constant IPD of ~130 ppb over the 20th century (Figure 2). Such constant IPD is unlikely, because $CH_4$ emissions are considered to have increased in the northern hemisphere for that period, which requires IPD to increase with time as discussed in the previous studies (Dlugokencky et al. 2003; Ghosh et al., 2015; Chandra et al., 2021). Accordingly, the BZ scenario appears a useful choice in firn modeling at Greenland sites, but it should be kept in mind that the use of $CH_4$ as a tuning tracer could lead to overfitting of the diffusivity profile.

A possibility to improve the reproducibility of the depth profile of $CH_4$ in the NGRIP firn could come from additional constraint to the diffusivity profile along with those currently made by the six trace gases ($CO_2$, $SF_6$, CFC-11, CFC-12, CFC-113 and $CH_3CCl_3$). In particular, it was indicated that $^{14}CO_2$, if available, would have strongly constrained the diffusivity profile and reduced uncertainty of the historical $CH_4$ trend reconstructed from the NGRIP firn. This study showed that the currently available firn data from Greenland (NGRIP and NEEM) are in better agreement with the historical $CH_4$ scenario prepared for the NEEM firn modelling (Buizert et al., 2012), then that for the CMIP6 experiments (Meinshausen et al., 2017). Since the latter scenario relies on the NEEM-S1 ice core data (Rhodes et al., 2013), this study highlighted inconsistency between the ice core and two sets of firn data in Greenland. Given that reconstruction of the $CH_4$ history from the deepest firn layers is challenging (in terms of the diffusivity versus history problem as shown in this study), future sampling and measurements of ice cores at a high-accumulation site in Greenland (where age of air occluded can be determined accurately) may be the only way to reconstruct the atmospheric $CH_4$ trend over the 20th century.

## Data Availability

The composition data of the NGRIP firn-air samples are available on the Arctic Data archive System (ADS) of National Institute of Polar Research (https://ads.nipr.ac.jp/dataset/A20210609-001). The NEEM firn-air data are available in the supplementary file of Buizert et al. (2012). The CMIP6 historical scenarios of the various trace gases used in this study are available via https://esgf-node.llnl.gov/search/input4mips/ as described in Meinshausen et al. (2017). Those of $^{14}CO_2$ are

available in the supplementary file of Graven et al. (2017). Our modeling data are available upon request
(umezawa.taku@nies.go.jp).

**Author Contribution**

TU, SS, KK and IO discussed on design of the study. KK, SA and TN conducted firn-air sampling at the NGRIP site. SS, KK, TU and TS analyzed the firn-air samples for trace gases. SJA set up the measurement system for the halocarbons. TU and SS made firn-air model simulations. TU analyzed the measurement/modeling data and prepared the manuscript with contributions
from all co-authors.

**Completing interests**

The authors declare that they have no conflict of interest.

**Acknowledgements**

We thank Morimasa Takata at Nagaoka University of Technology for assisting the sample collection, and Jakob Schwander
at University of Bern for collaboration at NGRIP. The NGRIP project was directed and organized by the Department of Geophysics at the Niels Bohr Institute for Astronomy, Physics and Geophysics, University of Copenhagen. It is supported by funding agencies in Denmark (SNF), Belgium (FNRS-CFB), France (IPEV and INSU/CNRS), Germany (AWI), Iceland (RannIs), Japan (MEXT), Sweden (SPRS), Switzerland (SNF) and the USA (NSF, Office of Polar Programs). We are grateful to the efforts for the measurement and modeling data from NOAA/ESRL/GML and the NEEM firn campaign, both of which
are made freely available. This work was supported by JSPS/MEXT (Japan) KAKENHI Grants-in-Aid for Young Scientists B (17K18342 for TU), Grants-in-Aid for Scientific Research on Innovative Areas (17H06320 for KK), Grant-in-Aid for Scientific Research (B) (20H04327 for IO) and the GRENE Arctic Climate Change Research Project (for SA). We thank the two anonymous referees for helpful comments to improve the manuscript.

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
