# Peer review of "Towards reconstructing the Arctic atmospheric methane history over the 20th century: measurement and modeling results for the NGRIP firn"

_Atmospheric Chemistry and Physics, 2021_

## Author Comment (AC1)

Response to Referee #1

We are very grateful to both referees for in-depth understanding of the present study and constructive suggestions. We believe that we have made our best efforts to consider questions/suggestions by both referees. We have made the following major corrections:

1. For our firn modeling, we have additionally used the CMIP6 scenarios for various trace gases (Meinshausen et al., 2017). The differences from those for the NEEM modeling (Buizert et al., 2012) are presented and discussed in a dedicated section of the revised manuscript. Both scenarios are examined for consistency with the NGRIP and NEEM firn data sets. The comparison of both scenarios highlights that they show a clear disagreement and produce a significant difference in reproducing the firn depth profiles for $CH_4$, but not for other trace gases.

2. We have made additional simulations for the NEEM firn with various diffusivity profiles in the same manner as the NGRIP firn. The iterative dating reconstructions of the historical $CH_4$ variations were also made from the NEEM firn. The reconstructions from both firn data are now presented and compared with the above scenarios.

3. Constraints from different trace gases have been evaluated by using the NEEM firn data. It turned out that $^{14}CO_2$ data play an important role in constraining firn diffusivity in the LIZ and thus reducing uncertainty in reconstruction.

4. We have concluded that, for $CH_4$, the Buizert et al. (2012) scenario is in better agreement with the two sets of firn data (NGRIP and NEEM) than the CMIP6 scenario. In addition, we point out that the former scenario is more consistent with the current understanding of the change in the interpolar difference (IPD) of atmospheric $CH_4$.

5. We have corrected all the figures accordingly and added necessary figures. Associated texts in many places are also modified.

Our responses to the Referee #1 are detailed below, where *referee's comments* and our responses are in different styles. The line, section and figure numbers in our responses are for the revised manuscript.

*General comments:*
*This paper uses firn air measurements from two Greenland sites (NEEM and NGRIP) to investigate the Arctic atmospheric methane history over the 20th century. A firn model is applied first to NEEM to demonstrate model performance, then to NGRIP to try to infer the Arctic methane atmospheric history.*

*A key assumption of this study is that the Arctic CH4 atmospheric history is uncertain but that the atmospheric histories of the other six gases (CO2, SF6, CFC-11, CFC-12, CFC-113 and CH3CCl3) are known with sufficient accuracy to constrain the firn model. More should be added to justify this assumption - I am not convinced. I do agree that it is a problem that the Arctic methane history is not known as well as has been assumed in previous firn model studies. However, the CO2 atmospheric history from Buizert et al is created in much the same way as CH4 (with an assumed offset from the SH ice core record), but has the added complication that CO2 has the possibility of elevated levels in NH firn due to in situ artefacts (e.g. mentioned in Buizert et al and elsewhere in the literature) - this could affect the ability of CO2 to constrain the firn model. The 14CO2 atmospheric history (relevant for NEEM, but unfortunately not measured at NGRIP) is probably quite reliable when it is based on atmospheric or tree ring measurements. The halocarbon histories are based on estimates of emissions, but these also have inherent uncertainty (the emissions themselves are due to*

*reported production/sales, assumed emission functions, and atmospheric lifetimes and therefore have uncertainties). I am not convinced that the atmospheric history of CH4 is significantly more uncertain than these other gases, I think all are known to some extent, but not perfectly.*

We agree with referee that the atmospheric histories of the many trace gases have significant uncertainties. To answer the referee's question about the degree of uncertainty for different gases, comparisons of the existing atmospheric synthetic histories by Buizert et al. (2012) and by Meinshausen et al. (2017) (hereafter referred to as the BZ and CMIP6 scenarios) are very helpful, which are now presented in the revised manuscript (section 3.2). We show in the revised manuscript that, for most trace gases, the atmospheric scenarios by the two independent studies are in good agreements, but the $CH_4$ scenarios are clearly inconsistent with each other (and small differences are obvious in some gases as well). We have made additional model simulations with the CMIP6 scenarios, which show that the modeled depth profiles differ significantly in $CH_4$ but with smaller magnitude in other trace gases. It is difficult to explicitly evaluate uncertainties of the atmospheric scenarios of individual trace gases, but the comparisons of the BZ and CMIP6 scenarios highlight considerable deficiency of our knowledge on the historical $CH_4$ variations in comparison to other trace gases.

*In addition to the question of how well atmospheric histories are known, it is also relevant to consider how well different gases can constrain firn diffusivity. Halocarbons measured in the deepest few firn samples at both NEEM and NGRIP are very close to zero, so do not provide a strong constraint on diffusivity in that region of the firn. As discussed at line 307, it is the region below about 74m at NGRIP that is used to infer the CH4 atmospheric reconstruction before 1980. The blue, orange and red lines in Figs 5 and 6 have a large spread below 74m for CH4 and CO2, but there is not a very large spread for the other gases, with the spread for some of these gases dropping rapidly to zero as depth increases. This shows that the modeled mole fraction profile for the CFCs, SF6 and CH3CCl3 in the deep firn is not very sensitive to the diffusivity profile, and consequently that the diffusivity profile is not as well constrained by these gases. It has been pointed out in previous studies that methane provides a strong constraint on diffusivity in the deep firn, but, as the authors note, only if the atmospheric history is well known, and unfortunately the authors are correct that it is not well known in the Arctic. CO2 would provide a similarly strong constraint on diffusivity, but I would suggest that the Arctic CO2 atmospheric history is also not well known and has the possibility of in situ artefacts in Arctic firn, as mentioned above. Thus, calibrating the firn model without CH4 for NGRIP, then expecting to reconstruct atmospheric CH4 is risky, and I believe the results show that it has not been successful (the model appears not to have been well constrained by the observations used).*

We thank the referee for this comment. To answer the referee's question, and as suggested by Referee #2, we have made additional model simulations for the NEEM firn (where $CH_4$ is excluded for diffusivity tuning as we made for NGRIP) to understand degree of constraints provided by different gases (section 5). The series of model simulations showed that large part of constraints to diffusivity in the deepest layers comes from the $^{14}CO_2$ data, which indicates that diffusivity in the NGRIP firn is relatively underconstrained in comparison to the NEEM firn, thereby subject to large uncertainty in reconstructing trace gas histories. Constraints from halocarbons (in addition to that from $CO_2$ as previously made by Ishijima et al., 2007) are found to be relatively weak in contradiction to our expectation at measurements. Confirming this, as well as suggested by Referee #2, we have decided to

conduct iterative dating reconstruction of $CH_4$ also for the NEEM firn in the same manner as the NGRIP firn (section 4.2 and Figure 10). We now therefore infer the historical $CH_4$ variations consistent with both firn data to compensate incompleteness of the NGRIP data set only.

*The most important contribution of this paper is questioning the assumption of a known Arctic atmospheric methane history for constraining firn models for Greenland firn sites. This has consequences both for calibrating firn models and for interpreting the CH4 north- south gradient in terms of emissions, as the authors discuss. However, as I have said, I believe the Arctic atmospheric histories for the other gases should be similarly questioned. I am not convinced that substantial conclusions have been reached in this study. The result that it is difficult to identify the atmospheric CH4 history that consistently reproduces the depth profiles of CH4 in NEEM and NGRIP firn is due to the fact that the firn model has not been adequately constrained by the other gases. The last 2 sentences of the abstract say that a consequence of this result is that the Artic CH4 history should be considered preliminary - it may be true that the methane history is not well known, however is not a consequence of that result. Rather, it is a prior assumption that has not changed as a result of the study.*

*While this study does highlight the deficiency that we don't know Arctic atmospheric CH4 well, in my opinion it doesn't go any way towards solving it. This makes me question the value of the study as it is currently presented.*

We thank the referee for the in-depth understanding and critical evaluation of this study. According to suggestions by both referees, we have examined additional model simulations and we believe that the present study made one step forward from the original manuscript to clarify the current best capability based on the two available firn data sets in Greenland (NGRIP and NEEM). The new series of simulations have confirmed difficulty in reconstructing the Arctic $CH_4$ history with small uncertainties. Albeit not a perfect success (as the referee points out), we consider that all of our efforts (measurements and modelings) and open issues are of importance and worth documenting so as to support future studies on Arctic firn and ice cores. We have concluded that for $CH_4$, the two Greenland firn data sets as well as the current understanding of the IPD prefer the BZ scenario over the CMIP6 scenario and that subtle $CH_4$ modeling for Arctic firn sites is still challenging, thereby suggesting that a new measurement of shallow ice cores may be the only way to significantly reduce uncertainty of the $CH_4$ history (the last paragraph in section 5). With the revision, these arguments are now better supported by the new model simulations. We are confident that our conclusions in this study are original and that this study has provided a path to better solve the question of reconstruction of the Arctic $CH_4$ history. We are very grateful to the referees for comments to improve our manuscript.

*Specific comments:*
*A conclusion in the abstract and at line 374 that "We find that, given the currently available firn air data sets from Greenland, reliable reconstruction of the Arctic CH4 mole fraction is possible only back to the mid 1970s" - atmospheric observations began around 1980, so this isn't much of a result. The title of the paper is 'Towards reconstructing the Arctic atmospheric methane history ...', but the study doesn't move very far towards that goal.*

According to the new additional model experiments, we believe that we have made one step forward towards the title of this study, which we hope the referee could agree. Please refer also to our response described earlier.

*Line 61 - "The NEEM-S1 data are notably higher than the ice core data after ~1850." The NEEM-S1 data after 1850 are fairly consistent with the Blunier et al 1993 data in Fig 1. The NEEM-S1 data are definitely higher than the Nakazawa data, but some of the Nakazawa data are lower than the SH Law Dome data which is unrealistic. Rhodes et al note that the uncertainty in absolute mole fraction of the NEEM-S1 data is about 6-9 ppb, and that that is a limitation to deducing the interpolar gradient, but perhaps the NEEM-S1 are our best chance at the moment to reconstruct NH methane between 1850 and 1945, seemingly better than the firn reconstruction presented here. The NEEM-S1 data were mentioned once in this study but otherwise dismissed (unfairly, in my opinion).*

We agree that the NEEM-S1 ice core data need to be more discussed, because it is the only available data that cover the period of interest at high resolution. In the revised manuscript, we did so by investigating the CMIP6 scenario which to a great extent relies on the NEEM-S1 data, and we found the following issues. First, our firn modeling results from NGRIP and NEEM sites indicated that the Arctic $CH_4$ history in line with the NEEM-S1 data (i.e. the CMIP6 scenario) cannot reconcile both NGRIP and NEEM profiles (Figures 8 and 9). Second, the NEEM-S1 data suggest $CH_4$ IPD of ~130 ppb around 1900, which is almost equal to that observed in the 1980s (Figure 2). Constant IPD between 1900 and 1980 is highly unlikely, because increase of $CH_4$ emissions for the intermediate period is considered to have occurred in the NH and it requires IPD to increase with time, as discussed in the previous studies (Dlugokencky et al., 2003; Ghosh et al., 2015; Chandra et al., 2021). Accordingly, we conclude that the CMIP6 $CH_4$ scenario (that is, the NEEM-S1 data) is consistent neither with the firn data nor the current our understanding on the atmospheric IPD. The data set may have an issue in data quality or dating, but such in-depth discussion is beyond the scope of this study. We have included above discussions in the revised manuscript.

*The strategy with prior and calibrated diffusivity profiles is not clear to me. For example: Line 177 - "The diffusivity profile optimised for the CIC model was tuned for our model" - what does this mean? Was the CIC profile used as a prior then improved by comparing to observations?*

We have reformulated the relevant sentences, after we made various diffusivity simulations also for the NEEM firn according to suggestion by Referee #2 (section 3.4). In the simulations presented in the original manuscript, we achieved good reproducibility with the diffusivity profile that was modified from the profile originally optimised for the CIC model. In the revised manuscript, we have used various diffusivity profiles including those prepared by likewise modifying the profiles optimised for other models.

*Line 182 - What diffusivity profile gave the RMSD value for NEEM of 0.83? Is this the same as the case shown in Fig 2?*

We have added the following sentence in the revised manuscript.

Line 244: "This RMSD value was achieved with an effective diffusivity profile that was prepared by modifying the profile originally optimised for the CIC (Centre for Ice and Climate) model at a certain range of depth."

*Line 197 - was that the initial diffusivity from equation 4 or Ishijima et al (2007)?*

In the manuscript, the initial diffusivity corresponds to that was used in Ishijima et al. (2007). This has been made clear as follows and throughout the revised manuscript.
Line 215: "In this study, the effective diffusivity profile prepared for the NGRIP firn by Ishijima et al. (2007) is referred to as the initial diffusivity and it was modified to improve the reproducibility of our newly measured trace gas profiles."

*Lines 198-203 - This paragraph is a little hard to follow, it became clearer as you read further, but could be improved. For example, line 198 "We examined the different sets of profiles" .. which different sets? (It becomes clearer, but is confusing at this point). Line 201 - "We prepared 100 different sets" - at this point the reader wonders how they are prepared, this also becomes clearer (page 12), but if this information was given when the steps are first discussed, it would improve readability.*

We have reorganized the explanations of the diffusivity profiles. The description of preparation of the 100 different sets of profiles have been now merged into section 3.3.

*Line 222 - which diffusivity profile was used in Fig 3? Eqn 4, Ishijima or a tuned profile? Why aren't the NGRIP results corresponding to the diffisivity profile giving RMSD=0.51 shown as a case (e.g. dashed black line) in Figs 4, 5, 6, 7 and 8? This would be good to see.*

The initial effective diffusivity profile from Ishiijma et al. (2007) was used here. The diffusivity profile giving the smallest RMSD was shown as part of red lines in the series of the figures. We do not think that the smallest RMSD case should be highlighted, because identification of a single case does not make a strong sense in this study where both diffusivity and atmospheric scenario have significant uncertainties.

*Fig 2 - It is difficult to see some of the observations, particularly CH4 in the deep firn. Could the observations be shown more clearly?*

We have made the symbols of the observations closed and larger in the figure (Figure 4).

*line 212 - the atmospheric history is not quite monotonic, so there could be more than one time with atmospheric mole fraction matching the mole fraction at the firn depth - how is that handled? Was the atmospheric history smoothed?*

We have added the following sentence in the revised manuscript.
Line 280: "It is noted that the smoothing spline curve applied to the BZ $CH_4$ scenario was used for calculation of the effective age, as the input scenario with seasonal variation (Figure 2) would not allow the effective age to be uniquely determined."

*Line 230 - at this point I'm already wondering what the modeled CH4 depth profile at NGRIP looks like with the Buizert et al atmospheric scenario in the model, but I need to wait....*

*The colored lines (red, orange, blue) cover many different diffusivity profiles, some of which don't fit the firn data well at all, particularly the blue cases. Is it worth showing the blue cases at all? At line 293, they are described as "less likely", but many of them simply do not fit the observations. Could the red group be split into two to highlight the really good cases? Do the better diffusivity profiles tend to fit all gases well, or do some profiles fit some of the gases well and others not so well, and vice versa (for groups of gases)?*

We hope to keep the light blue cases in the figures (Figures 3, 6, 8, 10 and 11) because it helps to see that the acceptable range was well narrowed after series of our simulations. As the referee also concerns, we have struggled between uncertainties of the diffusivity profile and the atmospheric scenario, and at this stage where neither can be tightly fixed, we do not wish to clearly highlight "really good cases" as it could lead misinterpretation of the present study. We hope the referee also understands that we try to present honest assessment of our current best use of the data. For the last question above, one "good" diffusivity profile tends to reproduce all gases well (except $CH_4$) as seen in the similarity of model-data differences for those gases in the initial simulations (Figure 5).

*line 297 "suggesting that the CH4 mole fraction may have been lower than the initial modeling scenario" - I am not convinced that this is a robust result. I am not convinced that the atmospheric scenarios are known more accurately for the other gases, or that they provide sufficient constraint on the model so that it can be used to infer the CH4 history, as discussed above.*

Please refer to our response described earlier.

*Line 309 - "The differences between the initial and corrected atmospheric CH4 scenario from these three deepest data are up to ~ 100ppb" - because the model is not well constrained by the other gases.*

Please refer to our response described earlier.

*Line 313 - "NGRIP firn data suggests decreased CH4 mole fraction from the 1950s to 1970s in any case, albeit with large uncertainty" - I do not believe this is a robust result, for reasons given above.*

Please refer to our response described earlier.

*Line 320-322 - if I understand this correctly, the CH4 history reconstructed from NGRIP gives a larger model-data difference at NEEM than the original history, is that not indicating an inconsistency?*

The referee is correct. The reconstructed $CH_4$ history from the NGRIP shows to some degree inconsistency with the NEEM firn data. To make fairer treatment, and as suggested by Referee #2, we have also made historical $CH_4$ reconstruction from the NEEM firn data (but $CH_4$ excluded from the diffusivity evaluation), and the reconstructions from both firn sites are compared in the revised manuscript (Figure 10).

---

## Author Comment (AC2)

Response to Referee #2

We are very grateful to both referees for in-depth understanding of the present study and constructive suggestions. We believe that we have made our best efforts to consider questions/suggestions by both referees. We have made the following major corrections:

1. For our firn modeling, we have additionally used the CMIP6 scenarios for various trace gases (Meinshausen et al., 2017). The differences from those for the NEEM modeling (Buizert et al., 2012) are presented and discussed in a dedicated section of the revised manuscript. Both scenarios are examined for consistency with the NGRIP and NEEM firn data sets. The comparison of both scenarios highlights that they show a clear disagreement and produce a significant difference in reproducing the firn depth profiles for $CH_4$, but not for other trace gases.

2. We have made additional simulations for the NEEM firn with various diffusivity profiles in the same manner as the NGRIP firn. The iterative dating reconstructions of the historical $CH_4$ variations were also made from the NEEM firn. The reconstructions from both firn data are now presented and compared with the above scenarios.

3. Constraints from different trace gases have been evaluated by using the NEEM firn data. It turned out that $^{14}CO_2$ data play an important role in constraining firn diffusivity in the LIZ and thus reducing uncertainty in reconstruction.

4. We have concluded that, for $CH_4$, the Buizert et al. (2012) scenario is in better agreement with the two sets of firn data (NGRIP and NEEM) than the CMIP6 scenario. In addition, we point out that the former scenario is more consistent with the current understanding of the change in the interpolar difference (IPD) of atmospheric $CH_4$.

5. We have corrected all the figures accordingly and added necessary figures. Associated texts in many places are also modified.

Our responses to the Referee #2 are detailed below, where *referee's comments* and our responses are in different styles. The line, section and figure numbers in our responses are for the revised manuscript.

*Umezawa et al. used a suite of gas measurements from NGRIP firn air (CO2, CH4, SF6, CH3CCl3, CFC-11, CFC-113, and CFC12) in combination with a firn model to reconstruct the atmospheric history of CH4 in the northern hemisphere (NH). Although the firn air samples were collected close to 20 years ago (in 2001), a great care has been taken to use state-of-the-art (or close to state-of-the-art) measurement techniques to achieve analytical precisions that are comparable or better than present-day modern atmospheric measurements. This is not a trivial merit and I think the authors should be commended. Following precedents set by previous studies of firn air (e.g., Rommelaere et al., 1997; Trudinger et al., 2002; Witrant et al., 2012; Buizert et al., 2012), Umezawa et al. used a forward gas transport firn model that takes in a "known" atmospheric history of a certain gas as an input and produce the expected mole fraction of that gas vs. depth profile in the open porosity of the firn. The difference between the expected mole fraction depth profile vs. measurements is then used to tune the "effective diffusivity" for this particular firn air sampling borehole (which is the Japanese firn sampling borehole at NGRIP).*

*A previous study by Buizert et al. (2012) set a precedent by including CH4 as part of the suite of gases used to tune the effective diffusivity at the NEEM ice core site. Buizert et al. (2012) achieved this by first making an educated guess about the "known" atmospheric history of CH4 in the NH. However, in this study Umezawa et al. challenge this assumption, treat the*

*NH atmospheric history of CH4 as an unknown, and only used the other six gas measurements (CO2, SF6, CH3CCl3, CFC-11, CFC-113, and CFC12) to tune the effective diffusivity profile for NGRIP. As a result, the atmospheric CH4 history reconstructed by Umezawa et al. has larger uncertainties; from this, Umezawa et al. argue that we cannot take the NH CH4 history for granted as a known variable to tune effective diffusivity profile for ice cores collected in the northern hemisphere and to certain extent, we also do not know the true atmospheric history of NH CH4 before the 1970s.*

*The main conclusion from of Umezawa et al. study (to which precision do we know the NH atmospheric history of CH4) is potentially an important one, so I would recommmend the manuscript for publication if the following comments are sufficiently addressed.*

We thank the referee for in-depth understanding of the present study.

*Major comments:*

*1. As reviewer #1 pointed out, it is not immediately clear whether the atmospheric histories for the other six gases outside of CH4 (CO2, SF6, CH3CCl3, CFC-11, CFC-113, and CFC12) used to tune the effective diffusivity profile are sufficiently known as well. Why focus on CH4 and not say, the uncertainty on NH CO2 history? I think a discussion or even a specific section addressing this question is warranted. Fortunately, given the current state-of-science knowledge, I think Umezawa et al. should be able justify their assumption in using CO2, SF6, CH3CCl3, and CFCs to tune effective diffusivity profile. Meinshausen et al. (2017) took a great care in synthesizing all available data from historical atmospheric measurements, firn and ice cores from several sites to best reconstruct the GHGs (including CO2, CH4, SF6, CH3CCl3 and the CFCs measured by Umezawa et al.) mole fraction, interhemispheric gradient, and seasonal variabilities for the purpose of CMIP6 model runs. This would be a great starting point. The justification for treating the NH histories of CO2, SF6, and the CFCs as "known" parameters, or at least better known parameters than NH CH4 history in my opinion should revolve around a discussion about the interpolar gradients of these gases (which are relatively small owing to their long atmospheric lifetimes), but I will leave the exact formulation of this argument to Umezawa et al.*

*I think a sensitivity analysis comparing what mole fraction should we expect in the open porosity of NGRIP firn if we put in NH vs. SH history from Meinshausen et al. (2017) for CO2, SF6, and the halocarbons is warranted to further drive the point home. I might be wrong, but I would expect the mole fraction vs. depth profiles for these suite of gases in the firn open porosity would not be as sensitive to NH vs. SH difference, at least relative to their respective measurement precisions compared to CH4 given their long atmospheric lifetimes and relatively low interhermispheric gradient. Given Umezawa et al. already had their forward firn model setup, hopefuly this does not require a lot of additional work. Furthermore, as a more general comment, I would also recommend Umezawa et al. to use gas histories from Meinshausen et al. (2017) for their overall firn gas transport and effective diffusivity tuning because the GHGs histories proposed by Meinshausen et al. (2017) represent more updated, better-educated "guesses" than the gas histories previously used by Buizert et al. (2012).*

We thank the referee for the constructive suggestions. In addition to the atmospheric scenarios by Buizert et al. (2012) (hereafter referred to as BZ scenario), we now also use the

synthetic atmospheric histories by Meinshausen et al. (2017) (hereafter referred to as CMIP6 scenario) for our firn transport model simulations. In the revised manuscript, we have added a section in which both scenarios are compared and their differences are described (section 3.2 and Figure 2). We highlight that, while the two scenarios show general agreements to each other for most trace gases, difference between the two scenarios is outstanding for $CH_4$. The $CH_4$ difference comes from the underlying datasets and assumptions for producing the respective synthetic data. The BZ scenario was produced by adding IPD to the Law-Dome-based Antarctic history, whereas the CMIP6 scenario employed the data from the NEEM-S1 ice core (Rhodes et al. 2013). While the BZ scenario assumed that IPD increased with the $CH_4$ growth rate (thus, with time) over the 20th century, the CMIP6 scenario suggests IPD to be almost constant and >100 ppb over the period. Except $CH_4$, IPDs of the other trace gases are sufficiently consistent with each other.

According to the referee's suggestion, we have made series of forward modelings for both NGRIP and NEEM firn sites using both historical scenarios (Figures 7, 8 and 9). We show in the revised manuscript that, the simulations with the CMIP6 scenario tend to overestimate the depth profiles of $CH_4$ to larger degree at both firn sites than those with the BZ scenario. For other six trace gases, the simulations using both histories do not produce significant differences.

We have also made the NGRIP firn simulations with the atmospheric scenarios for SH from Meinshausen et al. (2017) (Figures 5 and 8). We have found that input of the SH history for the NGRIP simulation resulted in significant differences even for the trace gases excluding $CH_4$. However, as the referee presumed, it has been found that relative differences between the simulations with the NH and SH scenarios are most pronounced for $CH_4$; while the differences for the other six gases are no more than 20 times the respective measurement precisions, that for $CH_4$ reaches about 40 times. This also emphasizes strong impact of IPD of $CH_4$ in comparison to other species.

In summary, we have concluded that (1) uncertainty of atmospheric history of $CH_4$ is manifest, while those of the other trace gases are relatively small so that they consistently reconcile the NGRIP and NEEM firn profiles, and (2) the atmospheric $CH_4$ history of CMIP6 is likely too high for the first half of the 20th century. Regarding the latter, we note that the increasing trend of IPD over the 20th century in the BZ scenario, in comparison to the constant IPD in the CMIP6 scenario, is more consistent with increasing anthropogenic emissions in the northern hemisphere suggested by earlier studies (Dlugokencky et al., 2003; Ghosh et al., 2015; Chandra et al., 2021). We have therefore corrected our argument. The BZ $CH_4$ scenario (Buizert et al., 2012) is the current best synthetic scenario, albeit large uncertainty and its use for tuning firn diffusivity unproven, and the firn air data are not consistent with the alternative CMIP6 scenario (Meinshausen et al., 2017).

*2. The suite of CFCs measurements (CFC-11, CFC-113, and CFC12), CH3CCl3 and SF6 do not provide good constraints for reconstructing effective diffusivity for the deep firn just because the concentration of these gases are all very low and close to zero. Usually, the gases that are most useful to reconstruct the effective diffusivity in this firn region are CH4, CO2, and 14CO2 due to their respective unique atmospheric histories. 14CO2 is especially useful as its atmospheric history can be validated from tree rings and historical atmospheric measurements. Furthermore, 14CO2 has a unique profile from the "bomb pulse" in the 1950s that provides a strong and unique constraint on the effective diffusivity. Unfortunately, 14CO2 measurements for NGRIP are not available. Because the CH4 history in*

*this study is treated as unknown, the effective diffusivity in the lower part of the NGRIP Japanese borehole presented by Umezawa et al. is almost solely constrained by CO2 data. This made me question whether the conclusion obtained by Umezawa et al. regarding how we cannot accurately reconstruct NH CH4 history from firn air samples is a unique problem pertaining to NGRIP (and its suite of gas measurements) or is it more general problem to other Greenland ice core sites as well. I don't think the current version of the manuscript sufficiently answer this question and additional work might be warranted to justify the conclusion put forward by Umezawa et al.*

*In particular, I think it would be especially useful to revisit the NEEM data from Buizert et al. (2012) with the same firn model and iterative dating algorithm presented in this study, but also excluding CH4 as part of the suite of gases to tune the effective diffusivity of the NEEM site. This would provide a more fair comparison rather than putting in the atmospheric history reconstruction from likely underconstrained NGRIP site into NEEM with a forward firn model. It would be interesting to see whether additional constraints from 14CO2 data at NEEM will allow for reconstruction of NH CH4 history with a better uncertainty and to what extent the uncertainty is better. For this experiment, I would recommend using the updated "known" 14CO2 history from Graven et al. (2017). Given Umezawa et al. already had their firn model tuned for the NEEM EU borehole as part of their model validation, I don't think this extra calculation would require significant amount of additional work.*

We thank the referee for the suggestions. As the referee points out, the constraints by halocarbons (CFCs and $CH_3CCl_3$) are relatively weak in contradiction to our expectations at measurements. From series of the simulations which we have made after the referee's suggestion, we found that large uncertainty in reconstructing the $CH_4$ history is a particularly pronounced problem for the NGRIP firn, and that the NEEM firn data set provides reconstruction with smaller uncertainty (section 5). This critical difference is ascribed to availability of the $^{14}CO_2$ data, by which we feel very regrettable for lack of $^{14}CO_2$ measurements for the NGRIP firn.

More specifically, according to the referee's suggestion, we have made forward model calculations also for the NEEM site. We have evaluated range of diffusivity profiles by trace gases excluding $CH_4$ and made reconstructions of historical $CH_4$ variations in the same manner as made for NGRIP (Figure 10). The result shows that the NGRIP-based reconstructions have larger uncertainties than the NEEM-based reconstructions. While the estimate of uncertainty of effective age at the two deepest depths (which roughly corresponds to the time period 1950–1970) exceeds 10 years, those at the corresponding NEEM depths (four deepest depths) are estimated to be less than 10 years (Figure 11). If $^{14}CO_2$ data were excluded for evaluation of the diffusivity profiles, we found that the uncertainties of effective age at the two deepest depths at NEEM would be increased to 15–20 years. Therefore, as the referee points out, the constraint from $^{14}CO_2$ data is strong, and the NGRIP reconstruction would have been different if $^{14}CO_2$ data were available. Relevant figures (Figures 10 and 11) and discussion have been added in the revised manuscript.

*3. I think the uncertainty analysis/discussion regarding the conclusion is a bit lacking. It is not immediately clear to me whether conclusion reached by Umezawa et al., that NH CH4 history in general should be considered preliminary and should not be used to tune effective diffusivity is sufficiently justified. From the study, it is clear that reconstructing CH4CH4 history from NGRIP firn air samples, when CH4 is excluded from the suite of gases used to*

*tune the effective diffusivity result in large uncertainties. But I think we know the NH CH4 history slightly better than just the reconstructed history from NGRIP firn air presented in this study.*

*Meinshausen et al. (2017) decided against providing uncertainties to the reconstruction of GHG histories that they did, arguing that the CMIP6 models would not have the computational resources to run multiple scenarios and sensitivity analysis from multiple GHG histories on top of the envisaged SSPs. I think an assessment about the uncertainty of historical CH4 reconstruction is very valuable and Umezawa et al. is in a unique position to take a first attempt at this. How about reconstructing NH CH4 history from NEEM (with its additional 14CO2 constraint) like discussed above, how about combining NGRIP, NEEM history inversion results to make a best-estimate of NH CH4 history and its uncertainties, and how about including CH4 in the suite of gases used for effective diffusivity tuning, but through iterative method starting first with larger uncertainty for the RMSD calculation to account for uncertainty in the CH4 history? There are still many avenues to explore beyond the reconstructed NH CH4 history from NGRIP firn samples before one can conclusively claim that we don't know the NH CH4 history to such a degree that it should not be included in the suite of gases used to tune effective diffusivity in firn profiles. I don't demand Umezawa et al. to do all of the above, as it might constitute a whole different study entirely, but a preliminary exploration on this and an honest assessment about how well can we reconstruct the NH CH4 history would significantly strenghten the manuscript and provide very valuable insights to the community.*

We thank the referee for the constructive suggestions. According to the referee's comment, we have additionally made reconstructions of historical $CH_4$ variations from the NEEM data (Figure 10). The $CH_4$ reconstructions from both NGRIP and NEEM are now combined and then compared with the BZ and CMIP6 scenarios. We realize that both scenarios were prepared with great care and used maximum number of data available at each time of the production, but they show significant differences in $CH_4$ for the early 20th century and earlier, as described above. While the BZ scenario follows the overlapping range of the reconstructions from NGRIP and NEEM back to around 1950, the CMIP6 scenario shows excursion to higher $CH_4$ mole fraction. Albeit large uncertainties of the reconstructions, we have concluded that the BZ $CH_4$ scenario better reconcile the currently available firn data from the NGRIP and NEEM sites.

We agree with the referee that it is of great value to assess uncertainty of the historical $CH_4$ scenario for climate modeling studies. However, such exact evaluation is still difficult because of the large uncertainties in reconstructing the $CH_4$ history from the firn data sets. A current possible conclusion is that the available NGIRP and NEEM firn data sets are in agreement with the BZ scenario better than the CMIP6 scenario. Considering that the CMIP6 scenario relies on the NEEM-S1 ice core data, this study highlight inconsistency between the ice core and two sets of firn data in Greenland. Rigorous evaluation/discussion of these available data sets is an important open question, but it is beyond the scope of this study.

*Minor comments:*
*I find that in general, the description about the firn gas transport models and the iterative method is very brief and might be bit hard to follow. The brevity is fine for the main manuscript, but the authors might want to consider a supplementary material where they*

*will have more room to describe the gas transport model, iterative methods, and especially additional data treatments. For example*
*Line 212 "Effective age at each sampling depth was calculated..." Several steps are clearly skipped here. It is not immediately clear to me, from the description of the model and equations above how one can determine the effective age at each sampling depth, as all the description before this line only pertains to the forward firn model. Did Umezawa et al. calculated a depth-age transfer function similar to Rommelaere et al. (1997) or through other means? Either way this needs to be elaborated.*

We have made descriptions of our modeling approach enriched in the revised manuscript (section 3), rather than adding a supplemental material that could make readers go back and forth.

The depth-age transfer function presented by Rommelaere et al. (1997) is indeed an interesting approach, and we have once examined similar age distributions for various trace gases at different depths at the NGRIP site (not shown). However, the effective age in this study was calculated in a simpler manner according to Trudinger et al. (2002), which was also used in Ishijima et al. (2007). The modeled $CH_4$ mole fraction at each sampling depth was compared to the input atmospheric scenario of the forward modeling, and the time at which the modeled $CH_4$ mole fraction agrees to the scenario was determined to be the effective age at the depth. As in Trudinger et al. (2002), the measurement data against the effective ages produce a renewed atmospheric scenario, which is then used for a forward modeling again. These steps are repeated to observe convergence of modeling results (iterative dating). We have reformulated the sentence as follows:
Line 276: "The iterative dating for $CH_4$ was performed as follows:
(*I*) Depth profile of $CH_4$ was calculated with the initial atmospheric $CH_4$ scenario.
(*II*) The modeled $CH_4$ mole fraction, calculated in step *I*, was compared to the input atmospheric $CH_4$ scenario, and effective age at each sampling depth was determined as the time when the modeled $CH_4$ agreed with a value in the atmospheric $CH_4$ scenario. It is noted that the smoothing spline curve applied to the BZ $CH_4$ scenario was used for calculation of the effective age, as the input scenario with seasonal variation (Figure 2) would not allow the effective age to be uniquely determined.
(*III*) A new atmospheric $CH_4$ scenario was constructed by assigning the observed $CH_4$ mole fraction, at each depth, to the effective age determined in step *II*. The observed $CH_4$ versus the effective age data set was interpolated by a smoothing spline function and it is considered as a revised atmospheric $CH_4$ scenario.
(*IX*) Depth profile of $CH_4$ was again calculated with the revised atmospheric $CH_4$ scenario constructed in step *III*.
(*X*) The above steps *II–IX* were repeated until the model-data difference converged within an acceptable range (typically after a few iterations) (Trudinger et al., 2002; Ishijima et al., 2007). In this study, we made five iterations for each modified diffusivity case as we confirmed sufficient convergence of the result."

*Fig.3. From the text it says "Figure 3 presents the initial simulations ..." Does this mean this is the initial effective diffusivity profile? It might also be beneficial to have the other effective diffusivity profiles like Fig.5 shown in Fig. 3.*

In the original manuscript, we intended to begin from the starting point of our modeling. The initial simulations in the original manuscript were made with the effective diffusivity profile

used for the previous study (Ishijima et al., 2007). After the reformulation of the manuscript according to the referees, we have now added the simulation results for the NGRIP firn with the atmospheric scenario for Antarctica (Figures 5 and 8), in order to highlight importance of IPD for different trace gases.

*There are several data treatment steps that is missing/the authors did not explain in sufficient details, or if the authors didn't do it, it is not well justified why they choose not to. For example, in their supplementary material Buizert et al. (2012) discussed how they added additional uncertainties for CO2 to account for possible in situ production and bubble close-off fractionation. In Buizert et al. (2012), uncertainty in atmospheric histories is accounted during the tuning of effective diffusivity by running the uncertainties through the forward model when the tuning of effective diffusivity is near complete to transfer the uncertainties from time domain to depth domain. I might miss it somewhere, but I think it is not immediately clear to me how the uncertainties of "known" atmospheric gases used to tune the effective diffusivity is treated in this study.*

We agree that our methodology of the data treatment, in particular for estimation of uncertainties, was insufficiently explained in the original manuscript. For the NGRIP firn, we only included the analytical precisions as the uncertainties in the RMSD evaluations, and did not include the possibility of in-situ production and close-off fractionation as done by Buizert et al. (2012) for the NEEM firn, because they are minor contributors with insufficient quantitative understandings (the atmospheric scenarios and analytical precisions are the two largest contributors to the total uncertainties). Therefore, the difference between the uncertainties of the NGRIP and NEEM firn reconstructions is largely due to the atmospheric scenarios. In theory, it would be possible to estimate the additional uncertainties for the NGRIP firn in the same manner as Buizert et al. (2012) and it would be a straightforward approach for comparison. However, in revising the manuscript according to the referee comments, we largely expand the simulations and now use the two independent atmospheric scenarios for all the trace gases, so that the uncertainties in the atmospheric scenarios are appreciably examined through the comparisons of the simulation results using the two scenarios. In addition, the complete comparison our results with those of Buizert et al. (2012) requires time-varying uncertainty estimates for the CMIP6 scenarios for all the gases, which is by itself a quite complex problem and beyond the scope of this study.

---

## Referee Report (RR1)

Umezawa et al. did significant work on improving the clarity and justifying their study. In particular, a lot of additional sensitivity analyses were conducted to clearly show why one needs to be careful when using CH4 as a "known" history to tune effective diffusivity in Greenland firn. Although the conclusion of this study is not very satisfactory (unfortunately we still don't know NH CH4 history that well), this conclusion is obtained through meticulous and comprehensive analyses of all available data and reconstruction techniques. Furthermore, this study provides a very conclusive result that shows that the NH CH4 history reconstruction from the "educated guess" used to tune NEEM diffusivity profiles (Buizert et al. 2012) is better than the one officially used for CMIP6 (Meinshausen et al. 2017). I recommend the manuscript for publication after some minor revisions (mostly stylistic).

**Major comments**

I think one aspect that is not sufficiently addressed is why the authors do not combine the result of NGRIP and NEEM (e.g., figure 10.c) to say something more definitive about NH CH4 history reconstruction inferred from both NGRIP and NEEM. It seems to me that there is a narrow band of CH4 history that satisfies both NGRIP and NEEM, and this at the moment should be considered our best NH CH4 history reconstruction. Umezawa et al. probably have good reasons on why they are reluctant to overinterpret this combined history, but I think the reasoning should be justified and discussed in the paper, otherwise readers like me are left wondering.

**Minor comments**

*Manuscript length, formatting, and redundancy*

I think the manuscript can be streamlined for clarity. It is clear that the writing on the new additions (seen in the track changes) is unfortunately not as concise as the original manuscript and sometimes somewhat redundant with other sections (often with the other new additions). Below, I provided some specific suggestions on parts that I personally think can be written more clearly, but this is more of a personal stylistic recommendation from me that in no way an assesment on the scientific merit of the paper. I also encourage the authors to look further and consider other parts of the manuscript can also be streamlined beyond my suggestions at their own discretion.

In its current state, the manuscript is unfortunately not very well formatted – it is hard to tell paragraphs apart because there is no indent or increased spacing between paragraphs. This made it quite challenging to review. I'm sure this will be fixed in the final proofs version. For example, the later ¾ of page 23 is a solid wall of text with no (obvious) break in paragraphs. If there is really no paragraph breaks, then I would recommend splitting the wall of text on page 23 and 24 into several paragraphs.

In the initial review, the two referees have asked for justification in why the authors are picking CH4 reconstruction above other trace gases. I personally think the authors have successfully shown throughout the paper that the motivation of this study is well justified. However, these explanations are scattered around throughout the paper. For example, one idea that is brought up in the introduction is the discrepancy between NEEM-S1 and other older ice cores, as well as BZ and CMIP6 reconstructions. Then not until page 16 that the authors run sensitivity analysis with regards to IPD to show that CH4 is uniquely uncertain. Finally, later on page 19 it is also discussed how the BZ and CMIP6 reconstructions result in different CH4 mole fraction over depth in NGRIP firn.

While I understand that all of the above are separate topics, I think the discussion to justify and show how CH4 is uniquely uncertain can be consolidated into one specific section, probably after Section 4.1 once the firn model and iterative methods are thoroughly explained. Then the authors can refer to this

section from early in the introduction when they describe the motivation of the study, why they think (and how they show) that CH4 is particularly uniquely underconstrained compared to other trace gases.

*More specific comments*
Page 3, figure 3: This figure has the NEEM-S1 scenario in red line overlapping directly with Arctic and Antarctic composite by Meinshausen et al. (2017) in green line. Please use a colorblind-friendly color palette to improve accessibility.

Page 4, line 77-88: Here the authors already refer to BZ and CMIP6 and discuss their discrepancy. This is (in some part) redundant with Section 3.2. where the authors formally designate the BZ and CMIP6 scenarios and also discuss their discrepancy. I think both parts can be combined.

Page 6, Eq. 1. I find that the authors are inconsistent in sometimes providing the units of all terms in the equations and sometimes they don't. For clarity, I would recommend giving the units to all terms in the equations (even the unitless term like porosity can be explicitly specified to be unitless).

Pg 10, line 222: "The various diffusivity profiles were constructed by modifying the original profiles at a certain range of depths in a stepwise manner."
I think this needs to be elaborated further. What certain range of depth? What is the range of the perturbation? "In a stepwise manner" how many steps? In figure 3 left panel, it is clear that the set of prepared diffusivity profiles is only perturbed between 50-65m.

Pg 11, line 247-248: "[…] profile that was prepared by modifying the profile originally optimized for the CIC (Centre for Ice and Climate) model at a certain range of depths."
Same questions as above. What certain depths and what is the range of perturbation away from the original $D_{eff}$ profile

Page 16, line 326-333. Here the authors are running a sensitivity analysis to show that CH4 is uniquely sensitive to IPD. I might be wrong, but I think here Figure 8 is cited before Figure 6 and 7. Furthermore, the authors need to state the purpose of this IPD sensitivity test before describing the result so that readers can have an idea of where the paper is going towards. In the current manuscript, the purpose of this sensitivity test is only just sort-of inferred after the result of the sensivity test is discussed in detail.

Page 17, figure 6. Here the black lines are clearly stated as initial diffusivities. The initial diffusivities are inconsequential but they are distracting because at first glance of the figure, presentation-wise the black lines seem like "best-fit" diffusivities. I would suggest the author to change the initial diffusivities into something that is more visually inconsequential (for example, dashed lines) and then use solid black lines as "best-fit" diffusivities if the authors are comfortable in showing the best-fit diffusivities on the plot.

Page 18, figure 7: Please add a note on the figure caption that this is NGRIP firn. I believe that all figures should be as standalone as possible, especially in a paper where there are a lot of similar figures like this one. The light blue of BZ and dark blue of CMIP6 are also hard to tell apart and I recommend the authors to use different and more distinct color palette to differentiate the two. Same with figure 9.

Page 22, line 440. I'm a bit confused on why the authors spend 5-6 lines to emphasize how in NGRIP reconstruction CH4 *can be* ~100 ppb off from the BZ scenario just to later say how highly unlikely the end-member CH4 reconstruction that is ~100 ppb off from the BZ scenario is. I understand that this is

to illustrate that NGRIP is really underconstrained when compared to NEEM, but this idea can be conveyed in a more concise manner and this analysis is only somewhat tangential to the main result.

, and this analysis is only tangential towards the main result

Page 25 line 509-onwards: I think the discussion about atmospheric d13C-CH4 reconstruction and how it is uncertain because uncertainty in CH4 history belong to the introduction as one of the motivation why we want to constrain NH CH4 history specifically, not the end of the discussion as it is only somewhat tangential to the result of this study.

*Other comments*

Pg 1, line 15: "[…] the early 1980s, and whilst CH4 measurements from Greenland ice cores …"
I would split the sentence into two. "[…] the early 1980s. *Although* CH4 measurements from Greenland ice cores …"

Pg 1, line 16-17: "In this study, we reconstruct the atmospheric CH4 for that period …"
In my opinion, for brevity this sentence is not necessary and can be combined with the following sentence.

Pg 1, line 17: "We use a data set of trace gases, measured from the air trapped in firn …" to "We use a suite of trace gas measurements from firn air … " again for brevity

Pg 1, line 24: "*reproduce*" to "*reproduces*"

Pg 1, line 26: "It is considered …".
This sentence is one of the main result of the paper, but unfortunately is written in passive voice and unclear manner. I would suggest writing something like "The atmospheric CH4 scenario used for NEEM firn air modeling is often considered the current best choice for Arctic CH4 history, but our study shows that until verified by further measurements it should not be used to tune firn models."

Pg 1, line 23: "*mid 1970s*" to "*mid-1970s*" with a hyphen (-)

Pg 4, line 68: "but their data are notably higher than *the* ice core data" to "higher than *other* ice core data" then cite the Eurocore and Site J paper.

Pg 4, line 86: "data set (red). Figure 1 shows that the two …"
I would just say "data set (red) and inconsistent with the Buizert et al. (2012) scenario." and then delete the following "Figure 1 shows that the two …" sentence because it prematurely described the result of the study, which is repeated multiple times later.

Pg 4, line 92: "using *the* iterative" to "using *an* iterative"

Pg 5, line 96-97: "[…] in May-June 2001. Accumulation, surface density, mean temperature …" to "[…] in May-June 2001. *Mean* accumulation, surface density, temperature …"

Pg 6, line 139: "occurs with depth, *which* stops at the top of LIZ" to "occurs with depth *and* stops at the top of LIZ"

Pg 6, line 150: "Namely, a trace gas flux in the firn" to "Namely, a trace gas flux *(F)* in the firn" add the *(F)*

Pg 7, line 168: "The closed porosity sc was calculated by empirical equation given by *Schawander* (1989)". First, Schwander's name is misspelled here. Also Schwander (1989) is unfortunately not an easily accessible chapter of a book (not available online through regular academic institution access). I presume the parameterizations Umezawa et al. refer to here are

scl (closed porosity) = s*exp*[75*($\rho$/$\rho$cod -1)] for $\rho$ <= $\rho$cod (close-off density)
scl = s for $\rho$ > $\rho$cod

I would recommend Umezawa et al. to just show the equations or cite other papers that show these equations together with Schwander et al. (1989)

Pg 8, line 195: "Therefore, CH4 is the only compound, with an available atmospheric history, which shows a clear disagreement, thus highlighting …".
To improve clarity I would just say something like "Thus CH4 is unique because currently it has two diverging synthetic histories that are only loosely constrained by observational data."

Pg 10, Eq. 4. "p" is obviously pressure but not formally defined within the paper.

Pg 13, line 284: "A new atmospheric CH4 scenario was constructed by assigning the observed CH4 mole fraction, at each depth" to "[…] at each *sample* depth"

Pg 19, line 385-386: "[…] produced enlarged overestimate in the LIZ (>63m) in the modeled profiles … ". "enlarged overestimate" is redundant. I would just say "[…] results in overestimation of CH4 mole fraction in the LIZ (>63m)"

Pg 22, line 440: "[…] suggested that the CH4 mole fractions over the period 1950-1980 could be *decreased in comparison to* the original BZ scenario." For clarity, I would recommend changing "*decreased in comparison to*" to "~100 ppb *lower*"

Pg 22, line 441: "The *decrease* of up to 100 ppb from the BZ […]" also for clarity, to "*The ~100 ppb lower CH4 mole fraction* from the BZ […]".

Pg 26, line 518: "but we regrettably report that the reconstruction of d13C-CH4 has not been possible despite our best modeling efforts (not shown)."
I think this sentence does not add anything to the manuscript, as neither the d13C-CH4 and dD-CH4 data nor "our best modeling efforts" are presented in the paper. I would suggest removing it entirely.

---

## Author Response (AR2)

**Response to Referee #1**

We are grateful to the referee for thorough re-review of our manuscript. Again, comments are very helpful for us to improve the manuscript. Our responses are detailed below, in correspondence to *referee's comment*.

*My concerns about the original version of the paper have generally been addressed. The most important part of this paper is questioning the assumption that NH CH4 is well known over the past century and can be used without considering uncertainties to tune firn diffusivity. For the paper's exploration of this topic, I believe it is worthy of publication. The limitations of the approach are now well spelled out.*

In the first review on the original manuscript, the referee provided important criticisms, which significantly contributed to our preparation of the revised version. We are very pleased to hear general agreement of the referee for the value of this study.

*Minor comments*
*line 16 - would it be more appropriate to say 'we try to reconstruct CH4 for that period'?*
According to suggestion by Referee #2, we have deleted the sentence.

*line 19 - the mention of NEEM at this point in the abstract understates the use of NEEM data in the paper. NEEM should be discussed more equally with NGRIP (details can be less, but at least mention the two sites, as it can be confusing as to how they are both used).*
According to this suggestion as well as comment from Referee #2, we have revised the sentences as follows:
"We newly report a data set of trace gases from the air trapped in firn (an intermediate stage between snow and glacial ice formation) collected at the NGRIP (North Greenland Ice Core Project) site in 2001. We also use a set of published firn-air data at the NEEM (North Greenland Eemian ice Drilling) site. The two Arctic firn air data sets are analyzed with a firn-air transport model, which translates historical variations to depth profiles of trace gases in firn."

*Line 49 - The data in the figure cover the last 300, not 200, years*
The caption has been corrected to 300 years.

*Fig 1 - the gray symbols for firn and ice core are hard to see on a printed copy*
According to this comment and suggestion by Referee #2, we have modified the color style of Figure 1.

*Line 72 - the Law Dome CH4 record has been updated by Rubino et al 2019 (https://doi.org/10.5194/essd-11-473-2019)*
We have updated the data in Figure 1 and cited Rubino et al. (2019).

*Line 77 and 81 - I don't understand use of the word 'synthetic' here, wouldn't reconstructions, scenarios or histories be better?*
We hope to keep the word here. We use the word to mean a data set that is to large extent supported by underlying assumptions as observational data are limited.

*Section 2 - NGRIP is described in this section, yet NEEM is barely mentioned. At least refer to Buizert et al in this section for NEEM details.*

At the end of this section, we have inserted the following paragraph.

"We also use a suit of trace gas measurement data from the NEEM site (77.45° N, 51.06° W). The firn air samples were collected in July 2008. The details of firn air sampling and gas measurements have been described by Buizert et al. (2012). The depth profile data of all the above trace gases ($CH_4$, $CO_2$, $SF_6$, CFC-11, CFC-12, CFC-113 and $CH_3CCl_3$) from the NEEM firn are used in this study as well as $^{14}CO_2$ data that are available from NEEM but not for NGRIP."

*Section 2 - Make it clear here that some gases (stating which) were measured and modelled previously by Ishijima, and which new gases are measured in this study.*

The following sentence has been added at the end of third paragraph of this section.

"Although not presented in this study, the firn-air samples were analyzed for nitrous oxide ($N_2O$) and its isotope ratios (Ishijima et al., 2007)."

*Section 3 - could mention depths of top of LIZ and deepest sampling at NEEM as well as NGRIP.*

These data have been added with the paper Buizert et al. (2012) cited.

*Line 169 - Schwander mis-spelt*

Corrected.

*line 182 - could quantify the maximum difference between the 2 scenarios*

The sentence has been corrected as follows:

"In contrast, the Arctic $CH_4$ histories by the two studies differ considerably with maximum difference of ~85 ppb around 1910 (Figure 2b)."

*Line 184 - could mention that the CMIP6 Antarctic histories are based on the Law Dome ice cores.*

The following sentence has been inserted:

Note that the CMIP6 scenario for the Antarctic latitude was constructed based on the Law Dome ice core data (see agreement in Figure 1).

*line 185 - "indicates almost constant values at ~ 130 ppb" - add "of IPD" i.e. "indicates almost constant values of IPD at ~ 130 ppb"*

Corrected.

*line 199 - the authors state that the two scenarios indicate that the other gases are at least better known than CH4 - could it be that they are based on the same underlying data? I'm not disagreeing, just be careful with the comparison.*

The referee is correct. We have corrected the sentence and added another sentence as follows:

"The scenarios of the individual trace gases have inherent uncertainties, but the comparisons of the two scenarios indicate that the data sources for other gases do not show inconsistent variations as seen in $CH_4$. It should be however noted that except $CO_2$, many trace gases lack observational data for the early 20th century, thereby both scenarios to a large extent being based on same data sources."

*line 209 - weren't simulated and observed profiles of more gases than CO2 used?*

No. Only $CO_2$ was used in the previous studies.

*line 210 - for clarity, state specifically the effective diffusivity as a function of depth. e.g. 'initial guess of the depth profile of effective diffusivity for CO2, Dinit(z)'*
Corrected.

*page 10 - say earlier in this paragraph that CH4 was not used here, e.g. around line 222*
We think that the explanation fits better at the original place. The sentence has been however modified to make it clearer that $CH_4$ was not used in the evaluation.

*line 260 - specify the trends are of CH4, and the two sets of scenarios are of CH4.*
This sentence explains trends and scenarios of other trace gases as well. For clarity, the first clause of the sentence has been corrected as:
"To represent atmospheric trends of different trace gases in the Arctic region, …"

*line 293 - for clarity, specify that the initial diffusivity was from Ishijima (fitted to which gases)*
The following sentence has been added:
"It is again noted that the initial diffusivity profile was tuned only for the depth profile of $CO_2$ (Ishijima et al., 2007)."

*line 298 - CFC-11 and CFC-113 have not increased monotonically, they have been decreasing since maxima in the 1980s and 1990s, respectively.*
We thank the referee for this correction. We have corrected the sentences as follows:
"It is known that, since the mid 20th century, the atmospheric mole fractions of the five trace gases ($CO_2$, $SF_6$, CFC-11, CFC-12 and CFC-113) have increased either monotonically or shown peak/slowed increase in the early 1990s (Sturrock et al., 2002; Martinerie et al., 2009). In contrast, $CH_3CCl_3$ has increased until the early 1990s and has rapidly decreased since then (Sturrock et al., 2002; Rigby et al., 2017), which is also observed in Figure 2."

*line 227 - could add 'shown by the dotted lines in Fig 5' at the end of the sentence that finished in line 227*
Corrected. We consider that this comment referred to line 327 of the former version of the manuscript.

*line 344 - could add 'for gases other than CH4' at the end of the sentence that finished in line 344*
Corrected.

*line 365 - "initial diffusivity profile of the BZ scenario" - doesn't make sense as written*
Corrected as "initial diffusivity profile and the BZ scenario".

*line 365 - it is important to know here whether Ishijima used CH4 to tune the model*
Ishijima et al. (2007) used only $CO_2$ and it has been clearly explained earlier.

*line 368 - 'also' rather than 'commonly'*
Corrected.

*line 372 - 'is used to the firn' - word missing? force or drive maybe?*

The word "force" has been added. The sentence has then been reformulated and moved section 4.5.

*Fig 9a - how significant is it that CO2 in the deep firn at NEEM is not particularly well modelled with either scenario?*

As seen in the figure, we do not find the difference between simulations with the different scenarios significant, and thus cannot identify which scenario better reproduces depth profile of $CO_2$.

*line 401 - when looking at the horizontal rows of circles, it may be hard to understand what the 'upper bounds' is, and may need better explanation*

The following words have been inserted: "…the upper bounds of the reconstructions (line connecting far-left red circles)…"

*line 422 - 'decreased CH4 mole fraction from the 1950s to 1970s' could be misunderstood as a decrease in time, it would be clearer to say 'lower CH4 mole fraction than BZ or CMIP6'. Similarly at lines 440 and 441, 'lower than' would be clearer than decrease*

Corrected.

*line 484 - Put \sigma_{age}(yr) into the figure caption, 'spread of effective age (sigma_{age}{yr}, maximum minus minimum)'*

This has been added.

*line 485 - specify in the caption that \sigma_{age}(yr) uses the right axis.*

This has been added.

*lines 492-494 - the way this sentence is written, it sounds like the result that consistent and accurate reconstruction is achievable only back to the mid 1970s is a universal result. Beginning the sentence with 'based on NGRIP and NEEM', doesn't make it clear enough that the situation could be quite different for other sites. I suggest reviewing this sentence.*

Following comments from both referees and revision at this time, we have modified the sentence as follows:

"Based on the currently best firn CH4 data from NGRIP and NEEM, we demonstrated that consistent reconstruction of the Arctic CH4 mole fraction is achievable back to the 1950s, but the uncertainty of reconstruction is still large (>30 ppb) for the 1950s to 1970s."

*line 503 rather than 'prefer', which sounds too informal, use 'are more consistent with', like in the abstract.*

Corrected.

**Response to Referee #2**

We are very grateful to the referee for thorough reading and comments/suggestions to improve the manuscript. Our responses are detailed below, in correspondence to *referee's comment*.

*Umezawa et al. did significant work on improving the clarity and justifying their study. In particular, a lot of additional sensitivity analyses were conducted to clearly show why one needs to be careful when using CH4 as a "known" history to tune effective diffusivity in Greenland firn. Although the conclusion of this study is not very satisfactory (unfortunately we still don't know NH CH4 history that well), this conclusion is obtained through meticulous and comprehensive analyses of all available data and reconstruction techniques. Furthermore, this study provides a very conclusive result that shows that the NH CH4 history reconstruction from the "educated guess" used to tune NEEM diffusivity profiles (Buizert et al. 2012) is better than the one officially used for CMIP6 (Meinshausen et al. 2017). I recommend the manuscript for publication after some minor revisions (mostly stylistic).*
We are very pleased to see such positive evaluation of our work after the last revision. The earlier suggestions by the referee had great contribution and significantly improved our manuscript.

*Major comments*
*I think one aspect that is not sufficiently addressed is why the authors do not combine the result of NGRIP and NEEM (e.g., figure 10.c) to say something more definitive about NH CH4 history reconstruction inferred from both NGRIP and NEEM. It seems to me that there is a narrow band of CH4 history that satisfies both NGRIP and NEEM, and this at the moment should be considered our best NH CH4 history reconstruction. Umezawa et al. probably have good reasons on why they are reluctant to overinterpret this combined history, but I think the reasoning should be justified and discussed in the paper, otherwise readers like me are left wondering.*
We thank the referee for this suggestion. We agree to the referee that the narrow band that satisfies both firn data corresponds to the likely range of the $CH_4$ history. Due to limitation of constraint to diffusivity in this study, we cannot identify the best reconstructed $CH_4$ history, but we consider that it would fall within the range. We have added a paragraph which explains our interpretation in the discussion section. The inserted paragraphs are as follows: "Whereas uncertainties in the reconstructions from the individual firn sites are large, Figure 11c could suggest a relatively narrow range of the $CH_4$ history that satisfies both NGRIP and NEEM reconstructions. Although the overlapping range of the two reconstruction is ~90 ppb in the 1970s, it is as small as ~30 ppb in the 1950s. This suggests that the combined NGRIP and NEEM firn data could provide a stronger constraint to the range of the $CH_4$ mole fraction e.g. 1185–1215 and 1225–1260 ppb in 1950 and 1955, respectively (Figure 10c inset). It is again noted that only the BZ scenario fall within these ranges for the period, suggesting that it is likely closer to the true atmospheric $CH_4$ history than the CMIP6 scenario for the mid 20th century."

*Minor comments*
*Manuscript length, formatting, and redundancy*
*I think the manuscript can be streamlined for clarity. It is clear that the writing on the new additions (seen in the track changes) is unfortunately not as concise as the original*

*manuscript and sometimes somewhat redundant with other sections (often with the other new additions). Below, I provided some specific suggestions on parts that I personally think can be written more clearly, but this is more of a personal stylistic recommendation from me that in no way an assesment on the scientific merit of the paper. I also encourage the authors to look further and consider other parts of the manuscript can also be streamlined beyond my suggestions at their own discretion.*

We thank the referee for the specific suggestions. We have made attempt to reformulate the manuscript so as to make it as concise as possible.

*In its current state, the manuscript is unfortunately not very well formatted – it is hard to tell paragraphs apart because there is no indent or increased spacing between paragraphs. This made it quite challenging to review. I'm sure this will be fixed in the final proofs version. For example, the later 3⁄4 of page 23 is a solid wall of text with no (obvious) break in paragraphs. If there is really no paragraph breaks, then I would recommend splitting the wall of text on page 23 and 24 into several paragraphs.*

We again thank the referee for thorough reading despite incomplete format. According to the comment, the corresponding paragraph has been split. We have also reformulated sections 4 and 5. We hope these reformulations have improved readability.

*In the initial review, the two referees have asked for justification in why the authors are picking CH4 reconstruction above other trace gases. I personally think the authors have successfully shown throughout the paper that the motivation of this study is well justified. However, these explanations are scattered around throughout the paper. For example, one idea that is brought up in the introduction is the discrepancy between NEEM-S1 and other older ice cores, as well as BZ and CMIP6 reconstructions. Then not until page 16 that the authors run sensitivity analysis with regards to IPD to show that CH4 is uniquely uncertain. Finally, later on page 19 it is also discussed how the BZ and CMIP6 reconstructions result in different CH4 mole fraction over depth in NGRIP firn.*

*While I understand that all of the above are separate topics, I think the discussion to justify and show how CH4 is uniquely uncertain can be consolidated into one specific section, probably after Section 4.1 once the firn model and iterative methods are thoroughly explained. Then the authors can refer to this section from early in the introduction when they describe the motivation of the study, why they think (and how they show) that CH4 is particularly uniquely underconstrained compared to other trace gases.*

We thank the referee for this suggestion. We agree that the corresponding discussion was not well incorporated in the revised manuscript after the first review. According to the referee's suggestion, we have inserted a sentence that mention to the $CH_4$'s uniqueness in introduction and added a paragraph that describes how series of modeling results demonstrated it before the section of $CH_4$ reconstruction (section 4.7).

*More specific comments*
*Page 3, figure 3: This figure has the NEEM-S1 scenario in red line overlapping directly with Arctic and Antarctic composite by Meinshausen et al. (2017) in green line. Please use a colorblind-friendly color palette to improve accessibility.*

We have modified the colors in the figure to make it more colorblind-friendly.

*Page 4, line 77-88: Here the authors already refer to BZ and CMIP6 and discuss their discrepancy. This is (in some part) redundant with Section 3.2. where the authors formally*

*designate the BZ and CMIP6 scenarios and also discuss their discrepancy. I think both parts can be combined.*

We have shortened descriptions about scenarios in this introduction part to refer section 3.2 for details.

*Page 6, Eq. 1. I find that the authors are inconsistent in sometimes providing the units of all terms in the equations and sometimes they don't. For clarity, I would recommend giving the units to all terms in the equations (even the unitless term like porosity can be explicitly specified to be unitless).*

We have added units for every term of equations at its first appearance.

*Pg 10, line 222: "The various diffusivity profiles were constructed by modifying the original profiles at a certain range of depths in a stepwise manner." I think this needs to be elaborated further. What certain range of depth? What is the range of the perturbation? "In a stepwise manner" how many steps? In figure 3 left panel, it is clear that the set of prepared diffusivity profiles is only perturbed between 50-65m.*

We have modified the sentence as follows:

"The various diffusivity profiles were constructed by modifying the original profiles at a certain range of depths in a stepwise empirical manner; depth range of the diffusivity profile key to improve reproducibility of trace gas depth profiles was first diagnosed and then the diffusivity was perturbed up and down in the depth range to the degree in which the corresponding simulated profiles do not deviate substantially."

*Pg 11, line 247-248: "[...] profile that was prepared by modifying the profile originally optimized for the CIC (Centre for Ice and Climate) model at a certain range of depths." Same questions as above. What certain depths and what is the range of perturbation away from the original Deff profile*

We hope the above explanation answers this question as well.

*Page 16, line 326-333. Here the authors are running a sensitivity analysis to show that CH4 is uniquely sensitive to IPD. I might be wrong, but I think here Figure 8 is cited before Figure 6 and 7. Furthermore, the authors need to state the purpose of this IPD sensitivity test before describing the result so that readers can have an idea of where the paper is going towards. In the current manuscript, the purpose of this sensitivity test is only just sort-of inferred after the result of the sensivity test is discussed in detail.*

We have reformulated the section to move the discussion about IPD later so that Figure 8 now appear in a correct order. Now the discussion is made in section 4.5. We have added motivation of the sensitivity test at the beginning of the section.

*Page 17, figure 6. Here the black lines are clearly stated as initial diffusivities. The initial diffusivities are inconsequential but they are distracting because at first glance of the figure, presentation-wise the black lines seem like "best-fit" diffusivities. I would suggest the author to change the initial diffusivities into something that is more visually inconsequential (for example, dashed lines) and then use solid black lines as "best-fit" diffusivities if the authors are comfortable in showing the best-fit diffusivities on the plot.*

According to the suggestion, we have modified the black lines from solid to dashed in Figure 6. However, we do not show "best-fit" result because our methodology in this study cannot specify one single profile that gives a best match e.g. the lowest RMSD.

*Page 18, figure 7: Please add a note on the figure caption that this is NGRIP firn. I believe that all figures should be as standalone as possible, especially in a paper where there are a lot of similar figures like this one. The light blue of BZ and dark blue of CMIP6 are also hard to tell apart and I recommend the authors to use different and more distinct color palette to differentiate the two. Same with figure 9.*

The caption has been modified to clearly state that this is NGRIP firn. We have also turned color of the CMIP6 scenarios blue to red, which has been applied for Figure 9, too.

*Page 22, line 440. I'm a bit confused on why the authors spend 5-6 lines to emphasize how in NGRIP reconstruction CH4 can be ~100 ppb off from the BZ scenario just to later say how highly unlikely the end-member CH4 reconstruction that is ~100 ppb off from the BZ scenario is. I understand that this is to illustrate that NGRIP is really underconstrained when compared to NEEM, but this idea can be conveyed in a more concise manner and this analysis is only somewhat tangential to the main result.*

We agree to the referee that the following sentences are indeed tangential. We have deleted those sentences because similar discussion appears later.

*Page 25 line 509-onwards: I think the discussion about atmospheric d13C-CH4 reconstruction and how it is uncertain because uncertainty in CH4 history belong to the introduction as one of the motivation why we want to constrain NH CH4 history specifically, not the end of the discussion as it is only somewhat tangential to the result of this study.*

We have deleted the paragraph from the discussion section and instead inserted shortened sentences in introduction to explain the motivation in terms of $\delta^{13}$C-CH$_4$.

*Other comments*
*Pg 1, line 15: "[...] the early 1980s, and whilst CH4 measurements from Greenland ice cores ..." I would split the sentence into two. "[...] the early 1980s. Although CH4 measurements from Greenland ice cores ..."*

Corrected.

*Pg 1, line 16-17: "In this study, we reconstruct the atmospheric CH4 for that period ..."*
*In my opinion, for brevity this sentence is not necessary and can be combined with the following sentence.*

We have deleted this sentence.

*Pg 1, line 17: "We use a data set of trace gases, measured from the air trapped in firn ..." to "We use a suite of trace gas measurements from firn air ... " again for brevity*

We have corrected the sentence as follows. The correction is slightly different from suggestion as a comment from Referee #1 has been also considered.

"We newly report a data set of trace gases from the air trapped in firn (an intermediate stage between snow and glacial ice formation) collected at the NGRIP (North Greenland Ice Core Project) site in 2001."

*Pg 1, line 24: "reproduce" to "reproduces"*
Corrected.

*Pg 1, line 26: "It is considered …". This sentence is one of the main result of the paper, but unfortunately is written in passive voice and unclear manner. I would suggest writing something like "The atmospheric CH4 scenario used for NEEM firn air modeling is often considered the current best choice for Arctic CH4 history, but our study shows that until verified by further measurements it should not be used to tune firn models."*

This sentence has been revised according to the suggestion.

*Pg 1, line 23: "mid 1970s" to "mid-1970s" with a hyphen (-)*

Corrected.

*Pg 4, line 68: "but their data are notably higher than the ice core data" to "higher than other ice core data" then cite the Eurocore and Site J paper.*

Corrected.

*Pg 4, line 86: "data set (red). Figure 1 shows that the two …"*
*I would just say "data set (red) and inconsistent with the Buizert et al. (2012) scenario." and then delete the following "Figure 1 shows that the two …" sentence because it prematurely described the result of the study, which is repeated multiple times later.*

Corrected as suggested.

*Pg 4, line 92: "using the iterative" to "using an iterative"*

Corrected.

*Pg 5, line 96-97: "[…] in May-June 2001. Accumulation, surface density, mean temperature …" to "[…] in May-June 2001. Mean accumulation, surface density, temperature …"*

Corrected.

*Pg 6, line 139: "occurs with depth, which stops at the top of LIZ" to "occurs with depth and stops at the top of LIZ"*

Corrected.

*Pg 6, line 150: "Namely, a trace gas flux in the firn" to "Namely, a trace gas flux (F) in the firn" add the (F)*

Corrected.

*Pg 7, line 168: "The closed porosity sc was calculated by empirical equation given by Schawander (1989)". First, Schwander's name is misspelled here. Also Schwander (1989) is unfortunately not an easily accessible chapter of a book (not available online through regular academic institution access). I presume the parameterizations Umezawa et al. refer to here are*
*scl (closed porosity) = s\*exp\*[75\*(ρ/ρcod -1)] for ρ <= ρcod (close-off density) scl = s for ρ > ρcod*
*I would recommend Umezawa et al. to just show the equations or cite other papers that show these equations together with Schwander et al. (1989)*

We thank the referee for this suggestion. We also found an error in our original manuscript about the prosity parametarization for NEEM, which has been corrected as follows. The previously published modeling on the NGRIP firn air (Sugawara et al. 2003; Ishijima et al.

2007) employed the porosity parameterization by Schwander (1989), and it is also the case in this study (e.g. the one correctly written by the referee). For NEEM, we employed the parameterization used by Buizert et al. (2012) i.e. the equation proposed by Goujon et al. (2003), for consistenvy with the previous NEEM firn air study and validation of our model. We confirmed that the simulations presented in this study were all performed accordingly, and added both equations in the text with references. We note that the choice of closed porosity parameterization should not have discernible impacts on the atmospheric reconstructions because the diffusivity profile is tuned for the respective cases.

*Pg 8, line 195: "Therefore, CH4 is the only compound, with an available atmospheric history, which shows a clear disagreement, thus highlighting ...". To improve clarity I would just say something like "Thus CH4 is unique because currently it has two diverging synthetic histories that are only loosely constrained by observational data."*
The sentence has been corrected according to the suggestion.

*Pg 10, Eq. 4. "p" is obviously pressure but not formally defined within the paper.*
"p" has been now defined.

*Pg 13, line 284: "A new atmospheric CH4 scenario was constructed by assigning the observed CH4 mole fraction, at each depth" to "[...] at each sample depth"*
Corrected.

*Pg 19, line 385-386: "[...] produced enlarged overestimate in the LIZ (>63m) in the modeled profiles ... ". "enlarged overestimate" is redundant. I would just say "[...] results in overestimation of CH4 mole fraction in the LIZ (>63m)"*
Corrected as suggested.

*Pg 22, line 440: "[...] suggested that the CH4 mole fractions over the period 1950-1980 could be decreased in comparison to the original BZ scenario." For clarity, I would recommend changing "decreased in comparison to" to "~100 ppb lower"*
Corrected.

*Pg 22, line 441: "The decrease of up to 100 ppb from the BZ [...]" also for clarity, to "The ~100 ppb lower CH4 mole fraction from the BZ [...]".*
This sentence has been deleted to incorporate the earlier comment.

*Pg 26, line 518: "but we regrettably report that the reconstruction of d13C-CH4 has not been possible despite our best modeling efforts (not shown)."I think this sentence does not add anything to the manuscript, as neither the d13C-CH4 and dD-CH4 data nor "our best modeling efforts" are presented in the paper. I would suggest removing it entirely.*
This sentence has been deleted according to the suggestion.